# Highly sensitive scent-detection of COVID-19 patients in vivo by trained dogs

**Omar Vesga**[1,2]*, **Maria Agudelo**[1,2], **Andrés F. Valencia-Jaramillo**[2,3], **Alejandro Mira-Montoya**[2,3], **Felipe Ossa-Ospina**[2,3,4], **Esteban Ocampo**[3], **Karl Čiuoderis**[5], **Laura Pérez**[5], **Andrés Cardona**[5], **Yudy Aguilar**[2], **Yuli Agudelo**[1], **Juan P. Hernández-Ortiz**[5,6], **Jorge E. Osorio**[5,6]

**1** Section of Infectious Diseases, Hospital Universitario San Vicente Fundación, Medellín, Colombia, **2** GRIPE, Universidad de Antioquia, Medellín, Colombia, **3** Colina K-9, La Ceja, Colombia, **4** Undergraduate School of Veterinary Medicine, Universidad de Antioquia, Medellín, Colombia, **5** Colombia/Wisconsin One-Health Consortium, Departamento de Materiales, Facultad de Minas, Universidad Nacional de Colombia, Sede Medellín, Colombia, **6** Department of Pathobiology, School of Veterinary Medicine, University of Wisconsin, Madison, WI, United States of America

* omar.vesga@udea.edu.co

**Data Availability Statement:** All relevant data are within the manuscript and its Supporting Information files.

## Abstract

Timely and accurate diagnostics are essential to fight the COVID-19 pandemic, but no test satisfies both conditions. Dogs can scent-identify the unique odors of volatile organic compounds generated during infection by interrogating specimens or, ideally, the body of a patient. After training 6 dogs to detect SARS-CoV-2 by scent in human respiratory secretions (in vitro diagnosis), we retrained 5 of them to search and find the infection by scenting the patient directly (in vivo screening). Then, efficacy trials were designed to compare the diagnostic performance of the dogs against that of the rRT-PCR in 848 human subjects: 269 hospitalized patients (COVID-19 prevalence 30.1%), 259 hospital staff (prevalence 2.7%), and 320 government employees (prevalence 1.25%). The limit of detection in vitro was lower than $10^{-12}$ copies ssRNA/mL. During in vivo efficacy experiments, our 5 dogs detected 92 COVID-19 positive patients among the 848 study subjects. The alert (lying down) was immediate, with 95.2% accuracy and high sensitivity (95.9%; 95% C.I. 93.6–97.4), specificity (95.1%; 94.4–95.8), positive predictive value (69.7%; 65.9–73.2), and negative predictive value (99.5%; 99.2–99.7) in relation to rRT-PCR. Seventy-five days after finishing in vivo efficacy experiments, a real-life study (in vivo effectiveness) was executed among the riders of the Metro System of Medellin, deploying the human-canine teams without previous training or announcement. Three dogs were used to examine the scent of 550 volunteers who agreed to participate, both in test with canines and in rRT-PCR testing. Negative predictive value remained at 99.0% (95% C.I. 98.3–99.4), but positive predictive value dropped to 28.2% (95% C.I. 21.1–36.7). Canine scent-detection in vivo is a highly accurate screening test for COVID-19, and it detects more than 99% of infected individuals independent of key variables, such as disease prevalence, time post-exposure, or presence of symptoms. Additional training is required to teach the dogs to ignore odoriferous contamination under real-life conditions.

**Funding:** OV received donations from Mauricio Palacio, Flor Saldarriaga, and LAS Sucesores SAS. JPHO received a donation from Grupo ISA. The funders had no role in study design, data collection and analysis, decision to publish, or preparation of the manuscript.

**Competing interests:** The authors have declared that no competing interests exist.

## Introduction

At the time of this writing, almost 33% of the world population has received at least one dose of a COVID-19 vaccine, but less than 1.5% of people in low-income countries belong to this group [1]. Under the most optimistic scenario, universal vaccine coverage is unlikely before 2023 [2]. Until this happens, early and accurate identification of people infected with SARS-CoV-2 is essential to prevent contagion [3]. Ideally, diagnostic tests must detect the pathogen in asymptomatic, pre-symptomatic and symptomatic patients [4]. The reference standard is the real-time reverse transcriptase-polymerase chain reaction (rRT-PCR); it is highly specific (~100%), but lacks sensitivity during the first 5 days post-exposure (0% on day 1, 33% on day 4, 62% on day 5), and its availability is limited [5]. Lateral flow antigen tests are less expensive, instrument-free, provide results faster than rRT-PCR, are quite sensitive during the pre-symptomatic period (84%-98%), and have ~100% specificity as well, but sensitivity is lost 5–7 days after exposure; all these make antigen detection more suitable to complement the insensitive period of the rRT-PCR [6]. Antibody tests are useless to prevent the dissemination of the virus, as they peak after the infectious period [7]. It was clearly demonstrated in several countries that early and massive testing, followed by immediate isolation in designated areas away from home and rigorous contact-tracing, were the only measures that effectively stopped the pandemic even before the first vaccine was available [8]. Quarantines provide time for health authorities to respond, but the benefit is doubtful [9], and the cost is catastrophic [10]. Vaccines offer the solution [1], but the time needed to immunize the world's population is more than enough for the virus to mutate and adapt [11]. Therefore, finding strategies to balance prevention against economic considerations is still an emergency.

Humans have been using dogs—*Canis lupus familiaris*—for scent-detection since the beginnings of domestication [12]. The great power of their sense of smell is exceedingly useful, and the first study of their olfactory capabilities was published more than 130 years ago by George J. Romanes [13], the research associate of Charles Darwin. Today, highly trained dogs are invaluable not only for their service [14], but also because their accuracy is superior to analytical instruments [15]. In the field of medical diagnosis, dogs are known to detect specific conditions [16], but most are anecdotal reports instead of formal protocols designed to validate a diagnostic test for clinical use [17]. However, at least one study demonstrated that appropriate training, coupled with strict adherence to the scientific method, lead to consistent diagnosis of *Clostridioides difficile* infection in humans [18], and, more recently, a comprehensive method was published validating canine diagnosis of two plant pathogens of international concern [19, 20]. Dogs detect and differentiate unique odors that result from the emission of volatile organic compounds (VOC) that constitute the "smell print" of the target [21]. In the case of SARS-CoV-2, several VOC have been found in the breath of COVID-19 patients [22, 23]. Since canines are inherently resistant to SARS-CoV-2 [24], and the virus cannot replicate in them or be transmitted from dogs to other mammals [25], it is not only safe but justifiable to study their efficacy and effectiveness in diagnosing COVID-19 by scent [26–29].

Our main objective was to determine the performance of scent-detection dogs as a screening tool in vivo for immediate detection of COVID-19 patients under a variety of circumstances [30]. The study was designed to address five research questions: 1) if working dogs belonging to breeds destined for non-scenting tasks would succeed as medical detectors (a positive result would increase significantly the canine population from which dogs could be selected); 2), the minimal number of COVID-19 patients required to train the dogs in vitro (such number must be enough for the dogs to make the inference that any human being with the same smell-print is a positive); 3) key diagnostic metrics (e.g. sensitivity and specificity) of scent detection in vitro and in vivo under controlled experimental conditions, i.e., screening

efficacy; 4) the canine limit of detection for SARS-CoV-2, quantified as number of copies of single stranded viral RNA per milliliter (ssRNA/mL); and 5) the real-life performance of scent detection dogs to detect SARS-CoV-2 in humans outside a hospital or office setting, i.e., screening effectiveness.

After proper training of six dogs, we compared canine diagnostic performance against the reference standard to determine relevant diagnostic metrics, including sensitivity (*SEN*), specificity (*SPC*), positive predictive value (*PPV*), negative predictive value (*NPV*), accuracy (*ACC*), and likelihood ratio (*LR*) of our dogs to detect by scent COVID-19 in vivo, i.e., by direct olfaction of the patient. The outcome was a very fast and cost-effective screening method for infection by SARS-CoV-2 in human patients.

## Materials and methods

Media data were uploaded to Figshare and are available at:
https://doi.org/10.6084/m9.figshare.14815848.v1

### Ethical statement

The study protocol was approved by the Ethical Committee for Human Research of *Hospital Universitario San Vicente Fundación* and the Animal Research Ethics Committee of Colina-K9. All human subjects read and signed their informed consent. We did not subject our dogs to any kind of pressure for training. We did not starve the dogs and did not need to make them obsessive for food, toys, games, or anything else. Since it is impossible to force a dog to do scent-work, our methods are exclusively positive, rewarding every correct response, and being indifferent to any mistake. Experiments with Syrian hamsters were carried out in strict accordance with the recommendations of the Guide for the Care and Use of Laboratory Animals of Universidad de Antioquia and the National Institutes of Health of the United States. No surgery was performed and animals were not subjected to any suffering or stress.

### Design

Fig 1 shows the training program and experimental design.

**Specimen collection for in vitro work.** This study was designed to determine the diagnostic performance of canines to detect, by olfaction, patients infected by SARS-CoV-2 in vitro and in vivo. The first step was to request written informed consent to aliquot, ultra-freeze (-70˚C), and thaw (for experimental use as needed) respiratory secretions from 12 COVID-19 patients admitted to three hospitals located in the metropolitan area of the Aburrá Valley, namely, *Clínica CES*, *Hospital Manuel Uribe Angel*, and *Hospital Universitario San Vicente Fundación* (Table 1).

**Dog training.** Using operant conditioning based on clicker-training and rewarding with food [31], we trained six canines to detect the odor print of SARS-CoV-2 in saliva and in the human body (Fig 2): four Belgian Shepherd Malinois (a herding breed), one first-generation cross Alaskan Malamute by Siberian Husky (a Nordic sled-dog), and one pit bull (a fighting breed).

Dog training was planned in three phases, each followed by its corresponding experimental work. Phase 1 (in vitro recognition) lasted 28 days during which we trained the dogs to recognize in vitro the scent-print of SARS-CoV-2 under a wide variety of environmental modifications that included, but were not limited to, time of the day, weather, training field, altitude above the level of the sea, distraction and hiding devices, age and temperature of the target samples, noise level and origin, distracting smells, training time, rewards, dog collars and leashes, etc. The only aspects of training that remained constant during phase 1 were the

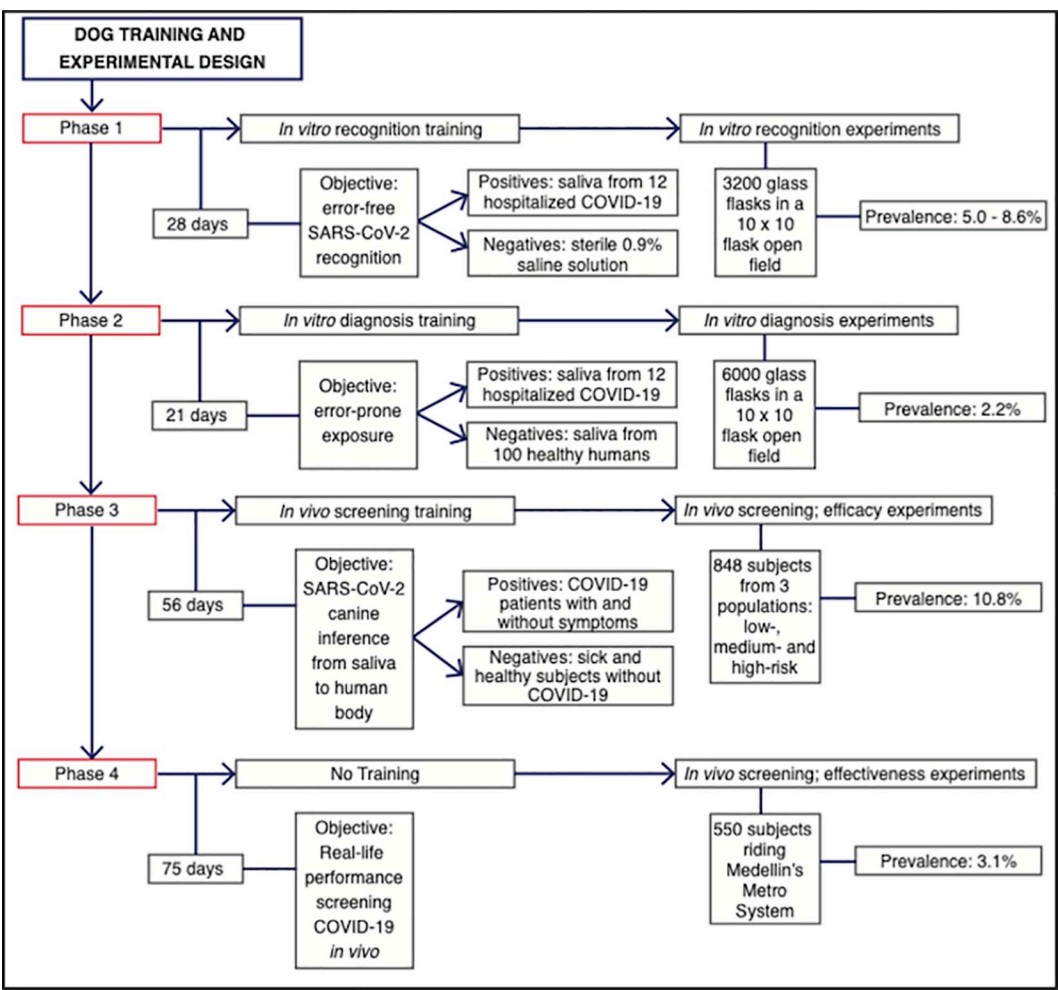

**Fig 1. Efficacy studies.** Flow chart depicting the order in which training phases and experimental design were conducted. The number of days after phases 1, 2, and 3 indicate the time employed training the dogs before running efficacy experiments; in the case of phase 4, the time without training before starting the effectiveness experiments. COVID-19 prevalence was set up as desired for in vitro experiments, introducing a more difficult scenario by minimizing prevalence during phase 2 (in vitro diagnosis). Prevalence during phases 3 and 4 (in vivo screening) was spontaneous, given by the pandemic epidemiology of the different human groups participating in the study.

positive specimens from the first three donors in Table 1 (one from each hospital), the use of 0.9% sterile saline solution as negative controls, the constant schedule of reinforcement, and the dog-trainer duos. We adhered to the errorless discrimination learning protocol developed by Terrace [32], which consists in presenting the animal a marked contrast between positive and negative stimuli during the foundations of training [33]. The "stimulus" is the problem presented to the dog, which was, for in vitro diagnosis, sterile saline solution (phase 1) or human saliva (phase 2), and for in vivo screening, the body of a person (phase 3). A stimulus can be positive if it leads to a reward for the dog (SARS-CoV-2), or negative, if it does not (controls). To recognize the scent-print of SARS-CoV-2, we trained the dogs to find their food (the reward) using their olfaction, always hiding with it a respiratory specimen from Patient 1 (the positive stimuli). The amount of food was diminished progressively until only the SARS-CoV-2 specimen was left in the hiding place, while the number of hides with saline increased in number. The director of training (AFVJ) pressed a clicker device to mark the correct behavior

**Table 1. Human subjects who provided specimens for in vitro training and experimentation (phases 1 and 2).**

| Patient # | Sex, Race | Age (y) | Specimen | Days Sick | SARS-CoV-2 rRT-PCR | Viral Load ($\log_{10}$ copies ssRNA/mL) |
|---|---|---|---|---|---|---|
| 1 | F, Hispanic | 74 | NPS & saliva | 12 | Positive | 5.42 |
| 2 | M, Hispanic | 55 | NPS & saliva | 10 | Positive | ND |
| 3 | F, Hispanic | 57 | NPS & saliva | 5 | Positive | 6.90 |
| 4 | M, Hispanic | 80 | TA | 16 | Positive | 5.15 |
| 5 | M, White | 29 | NPA | 10 | Positive | ND |
| 6 | M, White | 83 | TA | 10 | Positive | 5.09 |
| 7 | M, Hispanic | 34 | NPS | 3 | Positive | 10.2 |
| 8 | F, Hispanic | 27 | Sputum | 7 | Positive | ND |
| 9 | F, Hispanic | 26 | Sputum & saliva | 5 | Positive | ND |
| 10 | F, Hispanic | 61 | Sputum & saliva | 9 | Positive | 5.07 |
| 11 | M, Hispanic | 77 | Saliva | 8 | Positive | ND |
| 12 | M, Hispanic | 59 | TA | 13 | Positive | ND |
| 13–112 | 59F, 41M | R: 18–84 | Saliva | 0 | Negative | NA |

NPS: nasopharyngeal swab; TA: tracheal aspirate; NPA: nasopharyngeal aspirate; F: female; M: male; R: range; ND: not determined; NA: not applicable. The institutions in which the first 12 donors were hospitalized are not listed to prevent any risk of potential identification; the last 100 donors were ambulatory citizens.

(i.e., lying down to alert on the identification of the SARS-CoV-2 specimens), and the trainer immediately rewarded the dog, who had been conditioned beforehand to regard the click as the constant precedent (secondary reinforcer) to reward (primary reinforcer). Training sessions varied from 1 to 60 minutes, always followed by a resting period at least twice longer than the working time. It took one day for all dogs to understand that finding the SARS-CoV-2 specimen meant a prize for them, and that saline conveyed no reward. The following 27 days of phase 1, dogs were trained with respiratory secretions from Patients 1, 2 and 3 under the above-mentioned variations. The other 9 positive specimens (Patients 4–12) were reserved exclusively for experimentation, which only took place after the dogs had acquired the error-free skills necessary to identify SARS-CoV-2 with Patients 1–3 (defined as zero errors in a 10-sample field during 10 repetitions varying the target prevalence).

Phase 2 (in vitro diagnosis) went on for 21 days, keeping constant the positive stimuli (specimens from Patients 1, 2 and 3), but changing the negative stimuli for human saliva specimens donated by 100 human volunteers (Patients 13–112, Table 1). We exposed the dogs to a maximum of 10 specimens per training session (<10% positive stimulus plus >90% negative stimulus), reserving the 100-sample field for experimentation only. The donors of control samples were 100 healthy citizens belonging to the general population of the Aburrá Valley and its surrounding mountains; their saliva specimens were negative for SARS-CoV-2 by rRT-PCR the same day that we aliquoted and froze them at -70°C. Sample collection took place in March 2020, when the pandemic was just starting in Colombia and it was quite simple to find non-infected people. To prevent replication of the microbiota within each saliva sample, working specimens were thawed as needed, kept at 4°C between uses, and heat-sterilized before appropriate disposal. An illustration of the experimental field and the scent-detection work in vitro can be seen in S1 Video in S1 File (https://doi.org/10.6084/m9.figshare.14815848.v1).

Phase 3 of training (in vivo screening) took 56 days during which the dogs learned to identify COVID-19 patients by scenting the human body. Each canine was trained with 400 subjects who did not participate in the previous (or future) experiments, 100 hospitalized patients (COVID-19 prevalence: 40%) and 300 HCW (COVID-19 prevalence: 7%). Most of phase 3

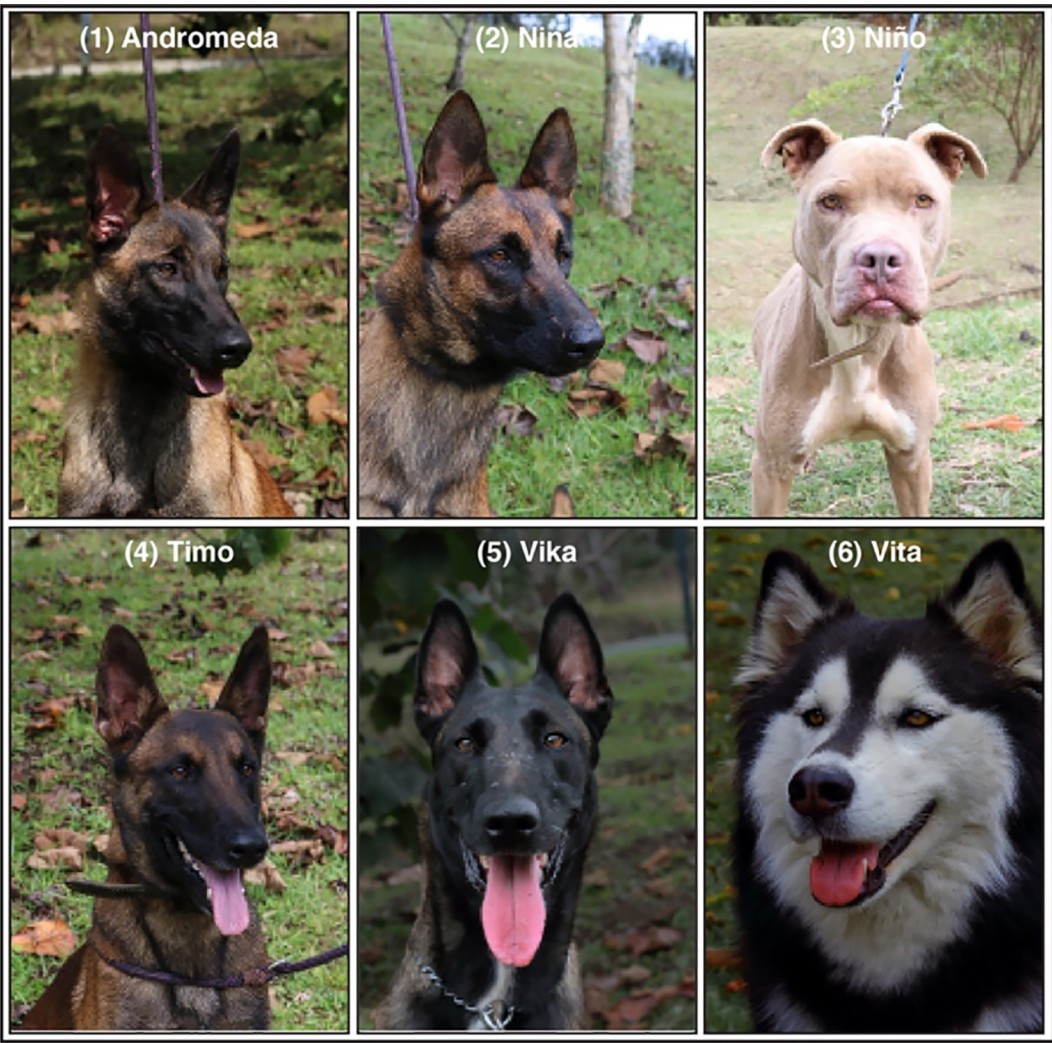

**Fig 2. Pictures and identification of the six dogs trained for the scent-detection of SARS-CoV-2.** (1) Andromeda, intact female, 6-mo, Belgian Malinois (BM). (2) Nina, intact female, 25-mo, BM, (3) Niño, castrated male, unknown age, American Pit Bull Terrier. (4) Timo, intact male, 31-mo, BM. (5) Vika, intact female, 36-mo, BM. (6) Vita, intact female, 36-mo, first generation Alaskan Malamute x Siberian Husky.

training time was dedicated to improving specificity and positive predictive value, because sensitivity and negative predictive value never represented an obstacle.

**Sample size.** Different sample sizes were used to train dogs in vitro and in vivo, because a much larger sample size is needed to validate in vitro scent detection [34]. Trial design is affected by prevalence and clinical severity of the disease to be diagnosed, because both variables have significant influence on *SEN*, *SPC*, *PPV*, *NPV*, *LR*, and *ACC* of the test under study [35].

In vitro, we set disease prevalence at random within specific ranges for each experiment, making it more difficult for the dogs as training progressed, 5% to 10% for phase 1, and 1% to 5% for phase 2. For a 5% prevalence rate and based on a target significance level of 0.05, at least 2140 samples were required to achieve a power greater than 80% in order to detect a change in sensitivity from 0.80 (null hypothesis, H0) to 0.90 (alternative hypothesis, Ha). To detect the same change in specificity, only 113 samples were required.

In vivo (phase 3), we recruited three populations based on their epidemiological risk for COVID-19: a high-risk group, consisting of patients admitted to *Hospital Universitario San Vicente Fundación* in Medellin, Colombia; an intermediate-risk group, consisting of health-care workers (HCW) at the same institution, and a low-risk group, consisting of officials working with the Governor of the Department of Antioquia. We anticipated a prevalence of 30%, 10% and 5% for each of these groups, requiring the participation of at least 63, 190, and 380 subjects to achieve the same targets as our in vitro testing, but with a change in sensitivity from 0.6 ($H_0$) to 0.9 ($H_a$); to detect the same change in specificity, only 27, 21 and 20 participants were required from each group, respectively. The expected average prevalence of COVID-19 for the whole sample was 10%, requiring at least 310 participants to achieve the same targets with a change in sensitivity from 0.7 ($H_0$) to 0.9 ($H_a$) and 34 participants to detect the same change in specificity [35]. To preserve power in case of lower prevalence, we aimed to include a larger number of subjects in all of the above-described scenarios. COVID-19 severity was expected to be proportional to risk, because prognosis for hospitalized patients and young people working in a government building should be at the extremes, while for HCW it should be in the middle. The sample sizes required (and attained) for sensitivity during experimental phases 1, 2 and 3 were 2140 (3200), 2140 (6000) and 310 (848), respectively; specificity requirements were much lower (S1 Table in S1 File).

## Experimentation after scent-detection training

**Blinding.** The first 60% of the experiments in phase 1 were unblinded (i.e., the handlers knew the position and number of positive stimuli in the field) to observe the behavioral cues offered by each dog during alerts on the target odor; the last 40% were blinded, as well as all experiments in phases 2 and 3, and the final scent effectiveness experiment. Except for phase 4, where diagnosis was unknown to everyone involved (scientists and participants), the director of training was always unblinded, and activated the clicker to inform the dog-trainer duos about every correct alert. In phase 4 he had to interpret the behavior of each dog to decide if a reward was in order.

**In vitro experiments.** For every experiment in vitro, the position of the samples in the field (1 to 100) and disease prevalence (1% to 10%) were randomized with a mobile phone app, and the dogs went through an open field arrangement of 10 x 10 samples (100) distanced 2 m in all directions. An illustration of the experimental field and the scent-detection work in vitro can be seen in S1 Video in S1 File (https://doi.org/10.6084/m9.figshare.14815848.v1).

Three kinds of 2-mL specimens were prepared under a biosafety class III laminar flow cabinet using 209 sterile, scent-free flasks: the control specimens consisted of 100 flasks with 0.9% sterile saline solution for phase 1, or saliva from 100 rRT-PCR-negative individuals for phase 2, while the experimental specimens consisted of 9 flasks with respiratory secretions from COVID-19 Patients 4–12 for phases 1 and 2. The positive specimens were diluted (1:1 volume) in 0.9% sterile saline solution to preserve the virus [36]. During in vitro experiments, each dog had to interrogate by scent a field with 100 flasks, the vast majority (90%-99%) containing negative stimuli; the rest would have the positive stimuli. After finishing a 100-flask field, the dog was offered abundant water and placed to rest in its individual kennel. Before the next search, each dog was scheduled to have an unrestricted play session and to take a long walk with its trainer. Once ready, the field was rearranged for a new experiment, changing at random the position of the specimens and the prevalence of COVID-19. Phase 2 differed from phase 1 only in that the negative stimulus was saliva from 100 healthy human volunteers and all experiments were blinded, i.e., the dog handlers did not know the position and number of positive specimens.

**In vivo experiments: Efficacy study.** For phase 3 in vivo screening, dogs could scent any part of the anatomy and were allowed to touch with their noses the body of the subjects. Because dogs usually sniffed the hands first, we instructed each participant to present the hands opened with palms facing the dog, as illustrated in S2 Video in S1 File (https://doi.org/10.6084/m9.figshare.14815848.v1). Phase 3 experiments took place at HUSVF (inpatients and HCW) and at the Governor's Building (government officials). Hospitalized patients were visited individually by the research team in their rooms or in the intensive care units. Government officials and HCW were screened in groups of up to 20 individuals in an open space in their respective institutions. After ending phase 3, we planned an additional experimental step without telling the training team about it, to determine real-life performance of the canine-human duos (phase 4).

**Phase 4, in vivo effectiveness.** These experiments were executed 75 days after the last experiment of phase 3 and entailed screening the general population riding the Metro System of Medellin (n = 550; 3 dogs of 3 breeds). Since we wanted to evaluate performance under real-life conditions, the human-canine teams were deployed to the field without previous announcement or environmental training, and participants were recruited on site without further delay. Since the research team had only three trainers, the effectiveness assay was limited to three dogs. As in phase 3, dogs were allowed to scent any part of the body, and each human volunteer provided a saliva sample for rRT-PCR once screened by the three dogs. As unique aspects of phase 4, dog performance was analyzed over time as they adapted to the Metro environment, and the director of training was informed of the experiments the day before, leaving rewards at his absolute discretion. In opposition to the method we had employed to train our dogs, it implied that some dog alerts might not be rewarded if interpreted as false positives by the training director.

## Limit of canine scent-detection

Freshly collected saliva specimens from four COVID-19 patients (unknown to the dogs) with viral loads ranging from 47 to 475 copies ssRNA/mL were serially diluted in sterile physiologic saline solution in 10-fold dilutions down to $1\times10^{-12}$ copies ssRNA/mL (i.e., 15 dilutions per specimen). Then, we randomized the dilutions from each patient by placing two COVID-19 dilutions along with 8 saline controls (10 flasks per row), and commanded every dog to search them until they finished the scent-interrogation of all 60 dilutions. The limit of detection (LOD) was the mean of the most diluted specimens that each dog was able to identify without failing a single one of the more concentrated dilutions. Since we did not determine the nature and relative concentrations of the VOC of COVID-19 patients, the only method available to quantify the LOD was the RNA concentration of SARS-CoV-2 per mL of saliva, which is very precise. It does not mean that RNA has odor, but provides a quantitative approach to the actual acuity of canine olfactory system to detect the scent-print of COVID-19.

## Dog-human teams biosafety: Evaluation of the SARS-CoV-2 containment devices

Besides strict adherence to the biosafety and patient isolation rules of HUSVF, we contrived two devices (D1 and D2) made with the fabric of a Dupont™ Tychem 2000 Coverall to prevent transmission of SAS-CoV-2 from speciments used in the study to pariticipating canines and their human trainers. The D1 device was used for scent-detection in saliva or respiratory specimens; it was a 130-mL glass flask with a metallic lid in which we perforated a 1 cm hole in the middle. The lid allowed a hermetic closure that remained intact after placing a 10x10 cm piece of Tychem 2000 between the bottle and its lid. The D2 device was used for the same

purpose but offered greater versatility than D1; it was a waterproof bag made of two 18x8 cm pieces of Tychem 2000, heat-sealed on all four sides (it contained inside a sterile gauze impregnated with the SARS-CoV-2 specimen).

In order to ascertain if any of the dog-trainer teams got infected by, or could have been at risk of exposure to SARS-CoV-2 during the project, we implemented two strategies. The first one was to run rRT-PCR tests on saliva specimens of dogs and trainers after ending phases 1 and 2. The second approach was to determine experimentally the efficiency of our containment devices in the Syrian hamster (*Mesocricetus auratus*) COVID-19 model (S1 Fig in S1 File). After an acclimatization period of 4 weeks, we exposed 15 animals (6 females and 9 males, 8 weeks old, outbred, immunocompetent) to SARS-CoV-2 over 4 days in groups of 3 hamsters of the same sex (2 experimental and 3 control groups), each contained in a HEPA filtered One System cage. Each of the two experimental groups had, inside their respective cage, one of the containment devices (D1 in group 1, D2 in group 2) protected by a metallic welded wire mesh enclosure that allowed hamsters to smell the device without touching it. Each of the three control groups had free access to an unprotected D1 flask (group A), a sterile gauze impregnated with a fresh specimen from a different COVID-19 patient (group B), or an unprotected D2 bag (group C). D1, D2 and the virus-impregnated gauze were replaced with fresh SARS-CoV-2 specimens every 12 hours in the 5 groups. All hamsters were sampled for rRT-PCR by saliva swabs before and after SARS-CoV-2 exposure.

## rRT-PCR assay and RNA quantification, RNA transcript standard generation, assay efficiency, and analytical sensitivity

The SARS-CoV-2 molecular diagnosis was conducted at the Genomic One Health Laboratory, Universidad Nacional de Colombia. Viral RNA was extracted from canine nasal and oropharyngeal swabs and human nasopharyngeal aspirates using the ZR viral extraction kit (Zymo Research) from a 140-μL volume of the specimens. Instructions provided by the manufacturer were followed and the sample was eluted into 20 μL. The CDC 2019-Novel Coronavirus Real-Time RT-PCR Diagnostic Panel (Integrated DNA Technologies) [37] and Berlin-Charité E gene protocol for SARS-CoV-2 [38] were used to detect virus nucleocapsid (N1 and N2) and Envelope genes, respectively. All rRT-PCR testing was done using Superscript III One-Step RT-PCR System with Platinum Taq Polymerase (Thermo Fisher Scientific). Each 25-μL reaction contained 12.5 μL of the reaction mix, 1 μL of enzyme mix, 0.5 μL of 5 μmol/L probe, 0.5 μL each of 20 μmol/L forward and reverse primers, 3.5 μL of nuclease-free water, and 5 μL of RNA. The amplification was done on an Applied Biosystems 7500 Fast Real-Time PCR Instrument (Thermo Fisher Scientific). Thermocycling conditions consisted of 15 min at 50°C for reverse transcription, 2 min at 94°C for activation of the Taq polymerase, and 40 cycles of 3 s at 94°C and 30 s at 55°C (N gene) or 58°C (R gene), and 3 min at 68°C for the final extension. SARS-CoV-2 assays were run simultaneously along with internal control genes for canine (glyceraldehyde-3-phosphate dehydrogenase-GAPDH) and human specimens (Ribonuclease P-RP) [39] to monitor nucleic acid extraction, sample quality, and presence of PCR reaction inhibitors [40]. To monitor assay performance, positive template controls and no-template controls were also incorporated in all runs. Biosafety precautions were followed during the workflow to minimize PCR contamination. For rRT-PCR qualitative detection, a threshold was set in the middle of the exponential amplification phase of the amplification results; a specimen was determined as positive for SARS-CoV-2 when all controls exhibited expected performance and assay amplification fluorescent curves crossed the threshold within 40 cycles ($C_T$ <40). For rRT-PCR quantitative detection of SARS-CoV-2 in human specimens, an analysis of copy number and linear regression of the RNA standard was used.

**Preparation of in vitro RNA Transcript as standard.** An in vitro RNA transcript of the SARS-CoV-2 envelope gene was generated as a standard for rRT-PCR quantitative detection in human specimens. Viral RNA from a positive clinical sample was used as an initial template for in vitro RNA transcription. cDNA was synthetized using SuperScript™ III First-Strand Synthesis System and random hexamers primer (Thermo Fisher, USA). Double-stranded DNA containing the 5′-T7 RNA polymerase promoter sequence for the SARS-CoV-2 complete E gene sequence, was obtained using DreamTaq Hot Start PCR Master Mix (Thermo fisher, USA) and E-Std-T7-Fwd (TAA TAC GAC TCA CTA TAG GGG CGT GCC TTT GTA AGC ACA A), and the E-Std-Rev (GGC AGG TCC TTG ATG TCA CA) primers [41]. The DNA was finally transcribed using the MEGAscript T7 Transcription Kit (Thermo Fisher Scientific). The RNA transcripts were purified with Ampure XP beads (Belckman Counter, USA) and quantified with a Qubit fluorometer using a Qubit RNA HS Assay Kit (Thermo Fisher Scientific). All commercial reagents were used according to manufacturer instructions.

**Assay efficiency and analytical sensitivity.** The in vitro RNA transcript standard was used to assess LOD and assay efficiency using a standard curve. Serial 10-fold dilutions of quantified in vitro RNA transcript were prepared in triplicate per dilution. The LOD for each assay was defined as the highest dilution of the transcript at which all replicates were positive. The efficiency (E) was estimated by linear regression of the standard curve using the equation $(E) = [10^{(1/slope)}] - 1$ [42]. The LOD and E of the SARS-CoV-2 assay were determined to warrant consistency with what has been previously demonstrated [43]. The intra- and inter-assay variability were also calculated using the in vitro RNA standard. To assess intra-assay variation, the RNA standard was used at 2 and 6 $\log_{10}$ copies/reaction by triplicate in a single assay. To assess inter-assay variation, the RNA standard was tested at 2 and 6 $\log_{10}$ copies/reaction by triplicate in two separate PCR assays. Mean, standard deviation, the coefficient of variation of the $C_T$ and copy numbers were also determined.

## Statistical analysis

Data input into 2x2 contingency tables generated the metrics *SEN*, *SPC*, *PPV*, *NPV* (mean and 95% confidence interval), *ACC*, and *LR*. Since pooling results from experiments with 100 specimens violates the independence assumption of the Fisher's exact test, we performed latent class analysis for in vitro data. Since no assumptions were violated in vivo, we applied the two-tailed Fisher's Exact Test to challenge the null hypothesis that the dogs detected COVID-19 by chance.

## Results

### Phase 1: In vitro recognition of SARS-CoV-2

The mean prevalence of SARS-CoV-2 for these experiments was 7.56% (range, 5.0%-8.6%), and the magnitude of all diagnostic metrics was very high (S2 Table in S1 File). The number of experiments varied for each dog because the required sample size (3200) was reached early and all of them recognized COVID-19 specimens with an accuracy >95.0%. To determine if diagnostic performance would improve by increasing prevalence to 20% (expected at the time of deployment), we set up an experiment with 40 flasks in a 10 x 4 field, allocating randomly 8 positive samples within 32 saline distractors. All six dogs identified correctly every sample without a single mistake, as expected from errorless training theory. With these results, dogs were ready for phase 2 training, designed for lower prevalence of positive samples (1%-5%) and greater difficulty in discriminating the positive from the negative stimulus (saliva from 100 non-COVID subjects instead of saline).

## Phase 2: In vitro diagnosis of SARS-CoV-2

Mean prevalence was 2.2%. Compared with phase 1, there was a significant improvement in the magnitude of all metrics for every dog (S3 Table in S1 File). As a group, the 6 dogs achieved *SEN* 95.5% (95% C.I. 90.4–97.9), *SPC* 99.6% (99.5–99.8), *PPV* 85.7% (79.2–90.5), *NPV* 99.9% (99.8–100), *ACC* 99.6%, and *LR* 267. The *PPV* improved 12 percentile points, while the *NPV* was close to perfection, thereby suggesting a very low probability that any of our dogs would miss a positive case in vitro (S2 Fig in S1 File).

## Phase 3: In vivo diagnosis of SARS-CoV-2 by direct body-scenting (efficacy trial)

One of the dogs (Vika) was excluded due to advanced pregnancy. Five dogs interrogated by scent 848 human subjects from three risk-groups: 269 hospitalized patients (high risk), 259 HCW (intermediate risk), and 320 government employees (low risk group). Demographics of the human participants are described in Table 2.

Before the canine scent-screening, we sampled the 848 participants to determine their COVID-19 status by molecular and antigen testing (Fig 3). SARS-CoV-2 infection was

**Table 2. Phase 3: In vivo screening, efficacy trial.** Demographic and clinical characteristics of 848 participants in the scent-detection experiments.

| Variable | | n (%) |
|---|---|---|
| **Sex** | All Participants | 848 (100) |
| | Females | 514 (60.6) |
| | Males | 334 (39.4) |
| **Age (years-old)** | Median | 56 |
| | Mean | 53 |
| | Youngest | 15 |
| | Oldest | 92 |
| **COVID-19 Prevalence** | All Participants | 92 of 848 (10.85) |
| | Government Employees | 4 of 320 (1.25) |
| | Health-Care Workers, HUSVF | 7 of 259 (2.70) |
| | Hospitalized Patients, HUSVF | 81 of 269 (30.1) |
| **COVID-19 Status** | SARS-CoV-2 positive | 92 (10.85) |
| | SARS-CoV-2 negative | 753 (88.8) |
| | SARS-CoV-2 indeterminate | 3 (0.35) |
| **Result by Reference Standard** | rRT-PCR positive | 41 (4.83) |
| | rRT-PCR negative | 753 (88.8) |
| | rRT-PCR indeterminate | 3 (0.35) |
| | Antigen positive | 51 (6.01) |
| | Antigen negative | 0 (0) |
| **Clinical Status at K9 Test** | COVID-19, asymptomatic | 18 (2.12) |
| | COVID-19, pre-symptomatic | 0 (0) |
| | COVID-19, symptomatic | 74 (8.73) |
| | Not COVID-19, but sick | 188 (22.2) |
| | Not COVID-19, healthy | 565 (66.6) |
| | Indeterminate, asymptomatic | 2 (0.24) |
| | Indeterminate, symptomatic | 1 (0.12) |

HUSVF: *Hospital Universitario San Vicente Fundación*.

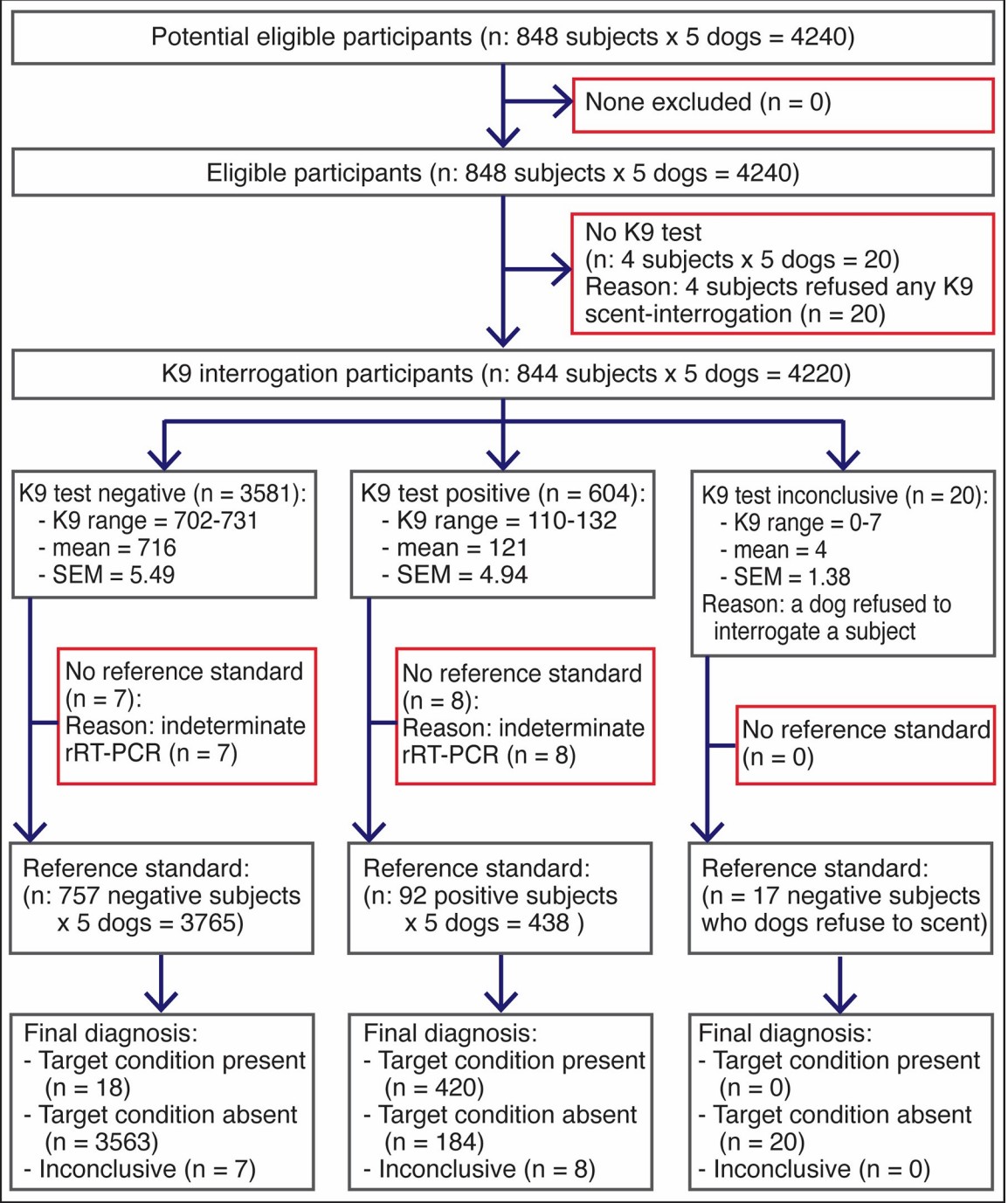

**Fig 3. Phase 3: In vivo screening (efficacy trial).** Diagram illustrating the flow of human participants in the third phase of the study.

confirmed in 92 patients (10.8%) and 753 (88.8%) tested negative. The other 3 (0.4%) were asymptomatic subjects with "indeterminate" rRT-PCR results after repeated testing; they were excluded from the analysis (Fig 4). The average cycle threshold ($C_T$) of the rRT-PCR positive patients was 32.0 (range, 20.6–38.8). COVID-19 was diagnosed by antigen test (Standard Q COVID-19 Ag Test, SD Biosensor) in 51 patients that had been admitted to the ER with acute respiratory distress, fever, sinus pain, cough, anosmia, or dysgeusia. Of 753 COVID-19 negative

**Fig 4. Phase 3: In vivo screening (efficacy trial).** Data analysis by risk group of all participants in experiments designed to determine performance metrics of the dogs during in vivo screening. Green, yellow, orange and purple cells contain true positives, false positives, false negatives, and true negatives, respectively. Cells not enhanced contain the number of participants with "indeterminate" rRT-PCR (3), subjects who declined K9 olfaction (4), and those rare occasions where the dogs refused to scent an individual, which happened 7 times with Andromeda and Nina and 2 times with Niño. Sensitivity could not be computed in the low risk group (NAN: not a number) because all 4 COVID-19 patients declined K9 scent-detection, resulting in 0 in two cells of the 2x2 contingency table and not significant P values in the two-tailed Fisher's Exact Test (enhanced in salmon color).

patients, 188 were hospitalized for other diseases that included respiratory conditions (23%, half had bacterial infections), malignancy (19%), autoimmunity (8%), coronary or peripheral atherosclerosis (7%), diabetes mellitus (6%), or chronic osteomyelitis (6%), and the rest had traumatic injuries, peritonitis, HIV, or cholangitis, among other pathologies. Of note, the dogs did not alert on any of the patients with respiratory diseases other than COVID-19. COVID-19 prevalence was 10.85% for the study population (92 of 848 subjects), distributed this way based on pre-test risk: 30.1% (81 of 269), 2.70% (7 of 259), and 1.25% (4 of 320) for the high, intermediate, and low-risk groups, respectively. As a group, the five dogs achieved *SEN* 95.9% (95% CI 93.6–97.4), *SPC* 95.1% (94.4–95.8), *PPV* 69.7% (65.9–73.2), *NPV* 99.5% (99.2–99.7), *ACC* 95.2%, and *LR* 19.6 (S4 Table in S1 File). Individual performance mirrored closely the group metrics (Fig 5). Four of 320 participants in the low-risk group had positive rRT-PCR tests, but these individuals declined canine scent-detection, producing zero values in two cells of the 2x2 contingency tables and precluding the computation of diagnostic metrics (Fig 4).

## Phase 4: In vivo screening of citizens riding the Metro System of Medellin (effectiveness assay)

The mass transit service of Medellin transports 1.5 million passengers every day. Without prior notification to the San Antonio station users or to trainers, three canines screened, over

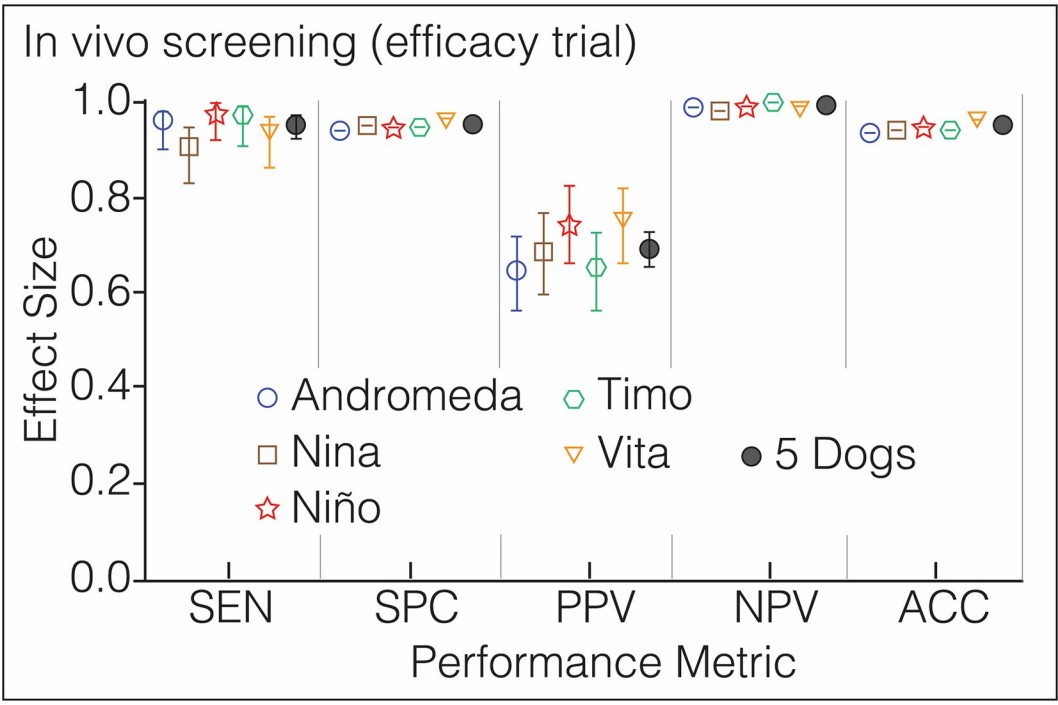

**Fig 5. Phase 3: In vivo screening (efficacy trial).** Performance metrics of 5 dogs screening for COVID-19 the patients and staff of *Hospital Universitario San Vicente Fundación* and the personnel working in the Office of the Governor of Antioquia; n = 848, global prevalence = 10.5%. Each symbol has a different color to ease visualization of the dogs. The vertical lines above and below the symbols represent the 95% confidence interval for each metric, which is contained within the symbol for *SPC*, *NPV* and *ACC*. Additional numeric data in S4 Table in S1 File.

two days, 550 individuals who also volunteered to provide saliva specimens for rRT-PCR testing. S2 Video in S1 File illustrates the level of difficulty of these crowded conditions for scent-detection work (https://doi.org/10.6084/m9.figshare.14815848.v1). Despite the environmental impact on the dog's concentration, they detected 17 COVID-19 cases with high *SPC* and *NPV*, 15 of them asymptomatic or pre-symptomatic. During the first 200 subjects, *SEN* and *PPV* dropped significantly in comparison with the efficacy trial (S5 Table in S1 File and Fig 6), but the dogs adjusted within 3 hours to the new environment and improved their performance until reaching a plateau (Fig 7). Table 3 shows the value for every diagnostic metric in each phase of the study.

## Limit of canine scent-detection

The LOD was determined in vitro using freshly collected saliva specimens from four COVID-19 patients new to the dogs. The moment of this assay coincided with the estrus cycle of several females, which caused the exclusion of the males from this experiment because both refused to work. The LOD for Andromeda, Nina, Vika, and Vita was lower than $2.61 \times 10^{-12}$ copies ssRNA/mL (S6 Table in S1 File), the equivalent of detecting a drop (0.05 mL) of any odorous substance dissolved in a volume of water greater than the capacity of 10.5 Olympic swimming pools ($2.6 \times 10^{10}$ mL).

## Biosafety of the canine and human team handling the virus

None of the dogs, their trainers, or the physician-scientists in charge of sampling and taking care of the patients contracted COVID-19 during this study. The rRT-PCR tests for

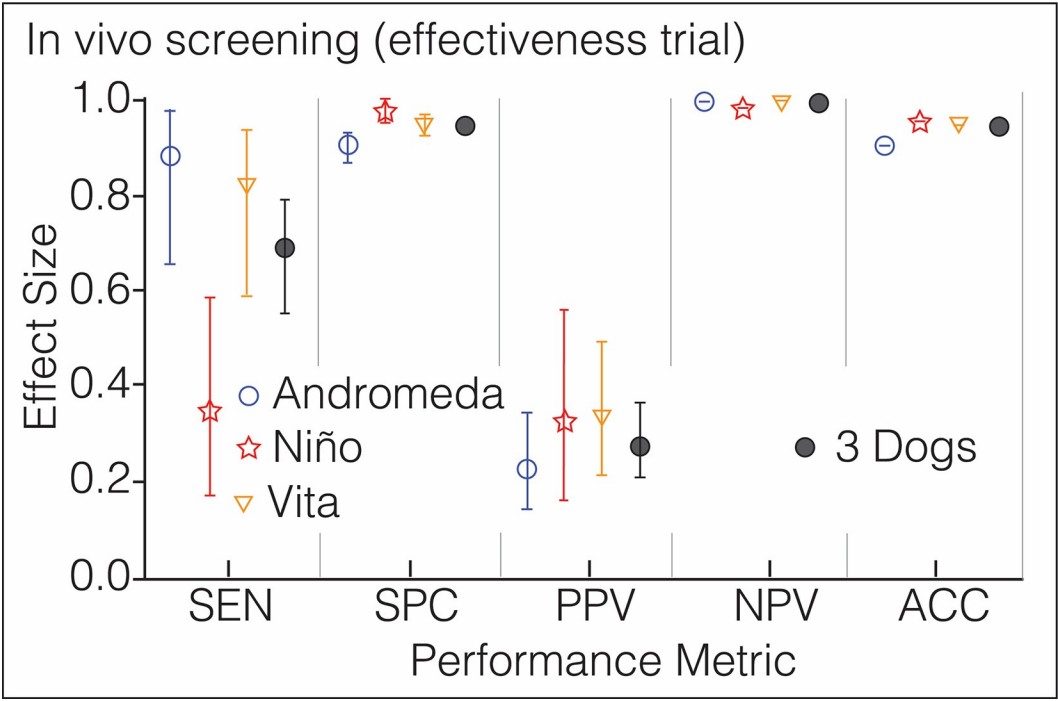

**Fig 6. Phase 4: In vivo screening (effectiveness assay).** Performance metrics of 3 dogs screening for COVID-19 the citizens riding the Metro System of Medellin; n 550, prevalence 3.1%. Each symbol has a different color to ease visualization of the dogs. The vertical lines above and below the symbols represent the 95% confidence interval for each metric, which is contained within the symbol for *NPV* and *ACC*. Additional numeric data in S5 Table in S1 File.

SARS-CoV-2 from canines and humans resulted negative twice, once after ending phase 2, and again after finishing phase 3 (S7 Table in S1 File). Experimental testing of the devices to contain SARS-CoV-2 showed that both worked as intended, allowing the scent to evaporate while holding the virus secured inside (Fig 8). The six hamsters in the experimental groups climbed and smelled the mesh-protected devices D1 (group 1) and D2 (group 2), but none acquired SARS-CoV-2. Hamsters in the control groups did climb on D1 but could not damage the Tychem 2000 fabric covering the flask, and none got infected (group A); did bite the Tychem of D2, and 1 animal was infected (group C); and played, licked, bit, nested, and slept in the gauze impregnated with SARS-CoV-2, and all three contracted SARS-CoV-2 (group B) (S8 Table in S1 File).

## Discussion

This study shows that canine scent-detection of COVID-19 is immediate, accurate, applicable anytime, and deployable anywhere as a diagnostic test in saliva or respiratory secretions, or as a screening tool in the patient directly. In any of those two roles, the dogs missed very few infected individuals, as demonstrated by *NPV* >99% in vitro and in vivo, and independently of the experimental design (in vivo efficacy and effectiveness trials). COVID-19 severity, ranging from asymptomatic to pre-symptomatic, sick and very sick patients, had no impact on performance. Prevalence from three populations of diverse levels of risk showed, as expected, that *PPV* went down when the presence of the disease in the population is very low, but *NPV* remained close to 100% across low and high prevalence. The errorless learning approach to training allowed generalization from only three specimens and prepared the dogs for in vitro

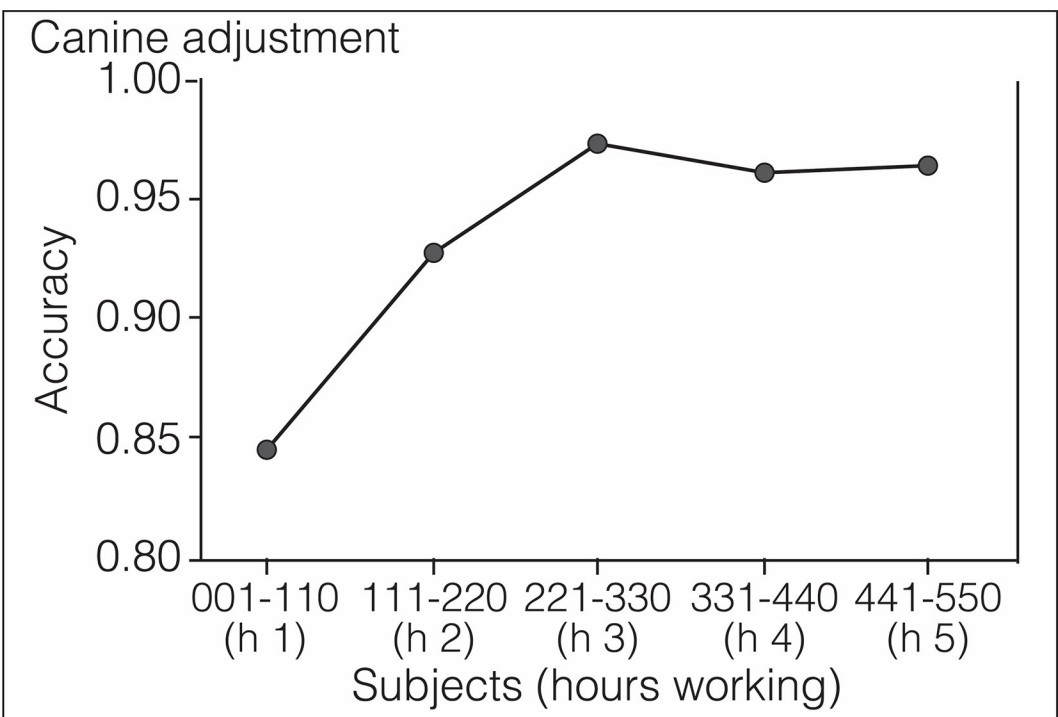

**Fig 7. Phase 4: In vivo screening (effectiveness assay).** Canine adjustment to a real-life situation. Accuracy started much lower under real-life conditions, but improved with time as the dogs adjusted to the new environment. Numbers labeling the abscissa represent the order in which subjects were screened by the dogs, divided in groups of 110 individuals. Screening each group took approximately one hour of work for the dogs.

diagnosis in improvised, open fields, making sophisticated and expensive equipment superfluous.

In vivo screening generated very encouraging results in both, efficacy and effectiveness trials, as the dogs detected more than 99% of the infected individuals spending <5 seconds per subject. Canine scenting of people has the potential risks of injury to patients (zero in this study), human refusal (four subjects), or dog refusal (16 cases among almost 6000 scent-screenings), but the advantages are overwhelming considering that this is the only screening test providing immediate identification and isolation of almost all infected subjects. It does not mean that in vivo screening is free of drawbacks: under real-life conditions, odor contamination caused a substantial increase in the false positive rate that drove the *PPV* down to 28%, which implies that most (72%) dog-positive subjects would have a negative rRT-PCR result, or that dogs produced 2.5 false positives for each true positive in the effectiveness assay. Although such error rate could still be acceptable for any screening test offering a very high *NPV* [44, 45], determining its cause could provide a method to solve the problem. One explanation was handlers rewarding some but not all alerts during the effectiveness assay but, in fact, rRT-PCR results showed that many correct alerts passed unrewarded; it confused the dogs and caused even more false negative alerts. The other reason is rather speculative, but based on experimental observations. The ultra-sensitive limit of detection suggests that at least a fraction of the false positives are actually pre-symptomatic COVID-19 patients. During training, three nurses whose rRT-PCR was negative were scored positive by the dogs, but 4–7 days later all three nurses had symptomatic COVID-19 with positive rRT-PCR. The dogs were accurate detecting those cases ahead of the molecular test. We also observed the dogs alerting spontaneously on

**Table 3. Summary of the results attained with six dogs trained to detect COVID-19 by scenting saliva and the body of human participants.**

| Diagnostic Metric | Phase 1: in vitro Recognition | | | Phase 2: in vitro Diagnosis | | | Phase 3: in vivo Screening (Efficacy Trial) | | | Effectiveness Assay (Metro System) | | |
|---|---|---|---|---|---|---|---|---|---|---|---|---|
| | Value | 95% C.I. | | Value | 95% C.I. | | Value | 95% C.I. | | Value | 95% C.I. | |
| SEN (%) | 88.8 | 84.3 | 92.2 | 95.5 | 90.4 | 97.9 | 95.9 | 93.6 | 97.4 | 68.6 | 55.0 | 79.7 |
| SPC (%) | 97.4 | 96.8 | 97.9 | 99.6 | 99.5 | 99.8 | 95.1 | 94.4 | 95.8 | 94.4 | 93.2 | 95.5 |
| PPV (%) | 73.9 | 68.6 | 78.6 | 85.7 | 79.2 | 90.5 | 69.7 | 65.9 | 73.2 | 28.2 | 21.1 | 36.7 |
| NPV (%) | 99.1 | 98.7 | 99.4 | 99.9 | 99.8 | 100 | 99.5 | 99.2 | 99.7 | 99.0 | 98.3 | 99.4 |
| ACC (%) | 96.8 | | | 99.6 | | | 95.2 | | | 93.6 | | |
| LR | 34.6 | | | 266.7 | | | 19.6 | | | 12.3 | | |
| P | <0.0001 | | | <0.0001 | | | <0.0001 | | | <0.0001 | | |
| Method | Latent Class Analysis | | | Latent Class Analysis | | | Fisher's Exact Test | | | Fisher's Exact Test | | |

the scientists that had touched any COVID-19 patient, or on the cell phones of nurses and physicians in care of COVID-19 patients. It suggests that our canines were making false alerts when detecting the scent-print of SARS-CoV-2 in contaminated individuals or in their belongings [46, 47].

Our data also provided answers to three other research questions. First, the limit of scent detection in vitro was lower than $10^{-12}$ copies ssRNA/mL, close to previous concentration thresholds determined with pure chemicals [48]. Second, all six dogs were successful as medical detectors, despite belonging to breeds not intended specifically for scent-detection. It supports recent data showing that, more than the breed, the best predictors of suitability for medical detection dogs are the levels of motivation, stamina, determination, resilience, and concentration ability of the individual dog [49, 50]. And third, only three COVID-19 patients sufficed for our dogs to recognize the scent-print of this particular disease in fresh saliva

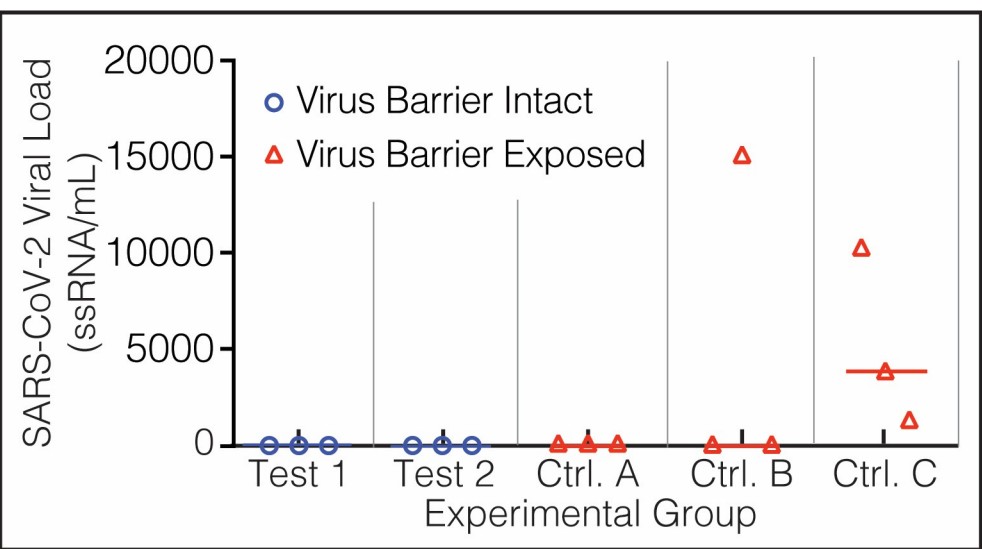

**Fig 8. Biosafety data.** Experimental evaluation of the devices used to contain SARS-CoV-2 specimens. After testing negative for SARS-CoV-2 in saliva, 5 groups of 3 golden Syrian hamsters each (*Mesocrisetus auratus*) were exposed during 4 days to SARS-CoV-2 directly (Group B, virus control) or enclosed in devices 1 and 2. Animals in Test groups 1 and 2 (blue circles) were allowed to sniff their devices but could not touch them, while those allocated to control groups A, B, and C (red triangles) had direct access to the containment fabric. The ordinate represents the viral load in saliva of each hamster after exposure to SARS-CoV-2 in 5 experimental groups.

specimens and in vivo. This process, called generalization, applies to learning theory, and, in reference to scent-detection, means that the canine ignores variations of the positive stimulus and indicates its source regardless of distracting odors [51]. Generalization after exposure to just three specimens is to be expected if errorless learning principles are the foundations of training [32]. It was demonstrated with pigeons 60 years ago [33], confirmed and expanded recently [52], and then proven with wildlife detection dogs [53]. Some experts believe that canines cannot generalize an odor when trained with specimens coming from a few patients, arguing that the dog memorizes the scent-print of the individual (the source) instead of the particular disease (the target odor) [54]. However, experimental evidence in favor of this hypothesis is scarce, and most citations refer to a work in which urine was employed as positive and negative stimulus during training [55]. Beyond the many variables specific to certain diseases and specimens that might be responsible for a greater level of difficulty for the dog, using the same type of secretion when the dogs are first trained for scent-detection does not favor errorless discrimination learning [32, 33, 52].

The magnitudes of the different diagnostic metrics in vitro and in vivo show that the dogs were looking for the scent-print of SARS-CoV-2 (their target) and not accessory odors memorized from Patients 1–3 or from the hospital environment. *SEN* and *SPC* measure the proficiency of the dogs to correctly discriminate between patients infected (*SEN*) or not (*SPC*) by SARS-CoV-2. In vitro (phase 2) and in vivo (phases 3 and 4), *SEN* was >95%, indicating that the dogs identified correctly almost all cases of COVID-19. Had they been alerting to odors other than their target odor, *SPC* would have been very low, and it was also >95%. While *SEN* and *SPC* refer to the index test (canine scent-detection), predictive values quantify the probability that the participants truly had COVID-19 or not, taking the reference standard as the truth (in reality, false positives and negatives also occur with rRT-PCR). In vitro and in vivo, *NPV* was >99%, while *PPV* was 85.7% in vitro (phase 2), 69.7% in vivo (phase 3), and 28.3% in the effectiveness assay (phase 4). Memory can be eliminated as an explanation for the low *PPV* under real-life conditions because, after 75 days without being exposed to a single person with COVID-19, the *NPV* remained above 99%. A better understanding of this and many other exceptional capabilities of our canines is provided by the abundant scientific data on dog behavior and cognition [56–59].

This study has some limitation that deserve attention. First, the lack of human coronavirus in our sample prevents their discrimination from SARS-CoV-2 or any non-human coronavirus. However, dogs did not alert on 43 hospitalized patients with respiratory conditions other than COVID-19, despite the fact that half of them had pneumonia caused by bacterial or viral pathogens like influenza virus. Second, the four COVID-19 subjects from the low-risk group opted out of the canine test. This precluded statistical calculations necessary to determine the different performance metrics under very low prevalence (1.25%). Data with HCW and Metro riders provide an approximation because prevalence was close (2.7% and 3.1%, respectively), suggesting that canine performance declined with prevalence. Although low prevalence is obviously not a problem with COVID-19, validation of this method might not be as successful with less frequent pathogens. Nonetheless, the excellent diagnostic performance in vitro under low prevalence (2.2%) indicates that improving the training method in vivo might overcome this particular barrier too. Finally, it should be noted that the advantages of using saliva instead of nasopharyngeal swabs are substantial and well supported [60, 61].

After our first preprint [62], at least four studies on canine scent-detection of SARS-CoV-2 in vitro have been formally published [63–67]. Despite substantial methodological differences with our work, results are reproducible. The main difference with those studies is that we chose to scent-interrogate the human body because of the many obvious advantages that such approach brings: results are immediate, can be obtained anywhere, do not require equipment,

and allow in situ separation of contagious individuals. The use of trained dogs as medical detectors was safe for the human participants during training and experimentation and effective regardless of the breed, a point of major importance considering that deployment would require the participation of many canines [68, 69], and the possibility of training dogs for real-time diagnosis of many other infectious diseases may help humanity be better prepared to confront the next pandemic [70]. These data suggest that well-trained dogs can be extremely helpful to guide societies through a safe re-opening of the economy and educational systems, while offering an efficient way to stop transmission. With improved training methods, canines could, in the near future, provide a sensitive and effective method to detect infectious diseases in a matter of seconds.

## Supporting information

**S1 File. Supporting information containing S1 and S2 Figs and Videos, as well as S1 through S8 Tables.**
(DOCX)

## Acknowledgments

We thank sincerely the patients, staff, and the members of the Board of Directors of *Hospital Universitario San Vicente Fundación*, the officers of the Governor of Antioquia, and the administrators and users of Medellin's Metro System for participating in the study. We are also indebted towards Dr. Hernán Alzate for allowing us to train his two dogs (Nina and Timo) for this project, and to Mr. Jesús González, Cosmovision journalist, for the excellent video footage. Our special gratitude goes to Dr. Tonie E. Rocke for her insightful review of the final version of the manuscript.

## Author Contributions

**Conceptualization:** Omar Vesga, Andrés F. Valencia-Jaramillo.

**Data curation:** Omar Vesga, Maria Agudelo, Karl Čiuoderis, Laura Pérez, Yudy Aguilar, Juan P. Hernández-Ortiz, Jorge E. Osorio.

**Formal analysis:** Omar Vesga, Karl Čiuoderis, Laura Pérez, Yudy Aguilar, Juan P. Hernández-Ortiz.

**Funding acquisition:** Omar Vesga, Juan P. Hernández-Ortiz, Jorge E. Osorio.

**Investigation:** Omar Vesga, Maria Agudelo, Andrés F. Valencia-Jaramillo, Alejandro Mira-Montoya, Felipe Ossa-Ospina, Esteban Ocampo, Karl Čiuoderis, Laura Pérez, Andrés Cardona, Yudy Aguilar, Juan P. Hernández-Ortiz.

**Methodology:** Omar Vesga, Maria Agudelo, Andrés F. Valencia-Jaramillo, Alejandro Mira-Montoya, Felipe Ossa-Ospina, Karl Čiuoderis, Laura Pérez, Yudy Aguilar, Juan P. Hernández-Ortiz, Jorge E. Osorio.

**Project administration:** Omar Vesga, Maria Agudelo, Yuli Agudelo, Jorge E. Osorio.

**Resources:** Omar Vesga, Maria Agudelo, Yuli Agudelo, Juan P. Hernández-Ortiz, Jorge E. Osorio.

**Software:** Omar Vesga, Karl Čiuoderis, Juan P. Hernández-Ortiz, Jorge E. Osorio.

**Supervision:** Omar Vesga, Maria Agudelo, Yuli Agudelo, Jorge E. Osorio.

**Validation:** Omar Vesga.

**Visualization:** Omar Vesga.

**Writing – original draft:** Omar Vesga.

**Writing – review & editing:** Omar Vesga, Maria Agudelo, Juan P. Hernández-Ortiz, Jorge E. Osorio.

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
