## [Decision Letter · Decision Letter 0]

23 Jun 2021

PONE-D-21-16873

Highly sensitive scent-detection of COVID-19 patients in vivo by trained dogs

PLOS ONE

Dear Dr. Vesga,

Thank you for submitting your manuscript to PLOS ONE. After careful consideration, we feel that it has merit but does not fully meet PLOS ONE’s publication criteria as it currently stands. Therefore, we invite you to submit a revised version of the manuscript that addresses the points raised during the review process.

I could obtain the comments from only one reviewer. He/she suggested that the manuscript was too long and difficult to follow. Thus, please rearrange the manuscript.

We look forward to receiving your revised manuscript.

Kind regards,

Etsuro Ito

Academic Editor

PLOS ONE

Journal Requirements:

1. Please ensure that your manuscript meets PLOS ONE's style requirements, including those for file naming. The PLOS ONE style templates can be found athttps://journals.plos.org/plosone/s/file?id=wjVg/PLOSOne_formatting_sample_main_body.pdf and https://journals.plos.org/plosone/s/file?id=ba62/PLOSOne_formatting_sample_title_authors_affiliations.pdf

Reviewers' comments:

Reviewer's Responses to Questions

**Comments to the Author**

1. Is the manuscript technically sound, and do the data support the conclusions?

Reviewer #1: Partly

2. Has the statistical analysis been performed appropriately and rigorously? 

Reviewer #1: Yes

3. Have the authors made all data underlying the findings in their manuscript fully available?

Reviewer #1: Yes

4. Is the manuscript presented in an intelligible fashion and written in standard English?

Reviewer #1: Yes

5. Review Comments to the Author

Reviewer #1: Since the outbreak of COVID-19 pandemic several peer-reviewed papers have been published, showing that dogs, even after a relatively short operant conditioning training, are able to discriminate odor samples collected from COVID-19 positive donors, from those collected from healthy controls. The studies demonstrated suprisingly high sensitivity and specificity >90% of COVID-19 detection by trained canines.

The general aim of these studies is to find a simple, inexpensive and high throughoutput screening method for detection of both symptomatic and asymptomatic humans infected with SARS-CoV-2 virus, to isolate such people to prevent further dissemination of the virus, even without being aware.

The reviewed paper generally supports previous findings, however, it includes some interesting issues that have not been addressed in previously published papers, as well as some novel methodological approaches. Therefore this paper has some merits that makes it worth publicating.

In my review I will focus chiefly on canine aspects of the study. Not being an epidemiologists, I feel not competent enough to evaluate the bio-safety and preventive measures to avoid contagion with SARS-CoV-2 during these experiments and to evaluate methods like rRT-PCR assay and RNA quantification to assess contagion hazard during collecting and handling of samples, as well as during in vitro and in vivo tests, as presented in the attached videos. However, in view of the latest news on new mutations of the virus, which are supposed to be more infectious that the previous variants, the procedure demonstrated on video S2 seems to neglect one of the main preventive measures that are recommended at least during the first phase of the pandemic, and namely distancing, avoiding direct contact e.g. by shaking hands, disinfection etc. Although the authors showed that the dogs and the persons involved in the experiments did not contracted the virus, demonstrating that nothing had happened at disobeying the preventive recommendations, does not mean that these recommendation can be generally disobeyed.

In my opinion, before accepting this manuscript for publication the structure of the submission should be improved, since it is chaotic, making the paper too long and difficult to follow. The authors divided the description of the methodology and results into main body of the paper and supplementary information. It is not clear to me why the authors placed a part of the description of the material and methodology in the main body of the submission and another part in the supporting information. The authors in the supporting info reiterate some details that were given in the main body of the submission. As supplementary info I would rather expect detailed raw data, video files (which are in fact attached, however, some short explanations to the details shown on the videos would be helpful), instead of a longer text with methodology, results, acknowledgements and references.

The authors listed 5 aims of their study

Aim #1 - Can dog breeds other than working breeds commonly used for olfactory detection, in this case pit-bull and nordic mix sled-dog (siberian husky x alaskan malamute), be trained and employed as medical detectors. However, using only one single dog of a breed does not tell much about ability of the breed to be efficiently trained for a specific olfactory task. In almost every dog breed there are individuals that can be successfully trained to perform any task with reasonable training workload and within reasonable time-frame, but it does not mean that a breed is generally recommended for the task. Therefore I suggest to discuss some limitation of this aim in the Discussion section.

Aim #2, assessing how is the lowest number of pattern odor donors for the training to achieve generalization of odor signature characteristic for COVID-19 in order to detect target odor in human population. It is generally recommended that dogs trained to detect diseases in humans should be trained on a number of pattern odor samples (high number of donors with a disease diagnosed). It is supposed that dogs could memorize individual components of odor samples (individual human odor) during training, if samples from too few donors are used, and then may have problems, when it comes to detection of odor samples collected from other donors. This issue is particularly important in COVID-19 detection due to high contagiousness if the SARS-CoV-2 virus and still not knowing exactly how people can contract the virus, especially with regards to new virus mutations. This would mean that the less COVID-19 positive donors from whom the samples have to be taken for the dog training, the lesser hazard to contract the virus. The authors domonstrated that pattern samples taken from only three donors, are sufficient to generalize the COVID-19 odor to the other nine people. The main question is, however, to what odor the dogs really alert. Previous studies showed that the dogs can be trained to alert to sweat samples (Grandjean et al. 2020, Angeletti et al. 2021) or to saliva or tracheobronchial secretions (Jendrny et al. 2020) collected from COVID-19 positive donors. As to the human sweat, it is commonly acknowledged that it contains individual odor component, genetically determined as well as influenced by bacterial decomposition. To my knowledge, no exact data are available as to the individual component of the odor of the saliva or tracheobronchial secretion. If the dogs almost perfectly generalize the sweat or saliva/trachoebronchial secretion odors of only three COVID-positive donors, this would mean that there is a defined, dominating odor of one or a single combination of few volatile organic compounds that masks the individual human odor. If ethyl butanoate was found as the most abundant VOC in the breath COVID-positive patients, (Chen et al. 2020), the question is if ethyl butanoate coud be considered a main candidate for a COVID-19 odor marker. Usually, however, it is a variety of combinations of VOCs in human odor samples, and as it was shown in studies on cancer odor markers, no single odor marker of the disease exist. The issue of the origin of the hypothetical COVID odor should be addressed in the Discussion section.

Aim #3, assessing the effects of sample numbers on the diagnostic metrics in vitro and in vivo under controlled experimental conditions, i.e., efficacy. Assessing such parameters as detection sensitivity, specificity and accuracy is a typical aim in such type of studies, allowing comparison and/or confirmation or throwing into question the results of other studies. The authors of the reviewed article include additional detection parameters like PPV, NPV and prevalence, which were not considered in any of the previous papers.

Aim #4 - Quantitative assessment of the detection threshold in terms of copies of single stranded viral RNA per milliliter (ssRNA/mL). This data would be helpful in collecting and preparing samples for canine trainig and testing. However, a question arises if the dogs alert to viral RNA as such, which is probably odorless, or to products of the changed cell metabolisms caused by the virus. Further question is if a given number of copies of ssRNA/ml is directly related to the amout of a single VOC or VOCs produced, to which the dogs probably alert.

Aim #5 - assessing real-life performance of the dogs in in vivo screening COVID positive passengers was an original and to my knowledge not addressed aim in previous studies on medical detection dogs trained to detect diseases in humans.

The in vitro part of the study uses experimental setup that differs from previously published studies both on cancer markers detection and COVID-19 detection by trained canines. The authors conducted the trials with dogs not indoors as it was the case in all other studies known to me, but outdoors, arranging 100 odor samples in 10 rows of 10 samples 2 m apart. This setup requires an area of 400 square meters, which may be not always availabe for experimentations. Secondly, conducting trials outdoors involves such hardly controllable confounding factors as distraction of dogs, weather conditions, uncontrollable migration of odor plume etc.

The videos attached to the suplementary info, show apparently a training phase under single-blind protocol meaning that the handler was blind to the position of the target sample in the lineup or to the COVID-positive person in the queue, but the experimenter who activated the clicker to give a signal to the handler that the alert was correct, was aware of the position of the target. While on the video S1 all alerts of the dog were rewarded, and the behavior of the dog, the handler and partly the experimenter can be seen, the video S2 was recorded probably for the purpose of the TV, incuding some shots from different positions that are impressive for the TV audience but does not document well the behavior of the dog, the handler and the experimenter . In the video S2 the dogs are sometimes rewarded after alert and sometimes not. Thus, it is not clear whether the video S2 shows a single blind training phase and alerts that are not rewarded were false, or it shows a real screening and the not rewarded alerts were true double blind trials. It would be useful to have some comments of the authors in the suplemenary info on particular alerts and rewarding of dogs recorded on the videos.

There is a basic difference to a real screening which is a true double-blind procedure, meaning that nobody knows if the alert was correct or false, and the dog should not be rewarded for a false alert. The dogs in this study are working on leash, which could be justified by a possibility to urge the dogs to work systematically i.e. sniffing all 100 in vitro samples and to sniff systematically people on the subway station. Although it cannot be seen that the handler gives overt cue to the dogs, and the dogs are well trained to work systematically, however, the leash is considered by many trainers a wired communication between handler and dog and is questionable in this type of detection. On the video S1 the experimenter with the clicker follows closely the dog in the scent lineup. Even if the dog does not seem to pay attention to the experimenter at this stage of deployment, it could be only a matter of time for the dog to learn to observe the experimenter to decipher when it should alert or not (the „Clever Hans effect”). Although theoretically there could be a variable reinforcement ratio, meaning that animals can be rewarded not after each correct behavioral response, but only after some, in practice the dogs, especially after a longer deployment, when not being rewarded frequently may try to earn a reward by using a trial-and-error strategy, which causes making more false alerts. The problem with detection dogs is that they work not to detect odors which have no biological relevance or rewarding values for canines, but the dogs rather work to find an opportunity to earn a reward, due to association between trained odor and a reward, that was created in the process of operant conditioning. Therefore the detection sensitivity and specificity may vary during the deployment period and should be systematically checked. At each of the training or deployment period an appropriate „success rate” should be settled to maintain dogs’ interest for work. While during the real screening test in the lineup of 100 samples it can be controlled how many samples are of known status (positive and negative) and how many samples are of unknown status, when testing people in vivo in a queue, it is hardly to controll who of them are true positive, true negative to reward/not reward the dog for a correct response, unless such persons diagnosed previously are available for the trials. This issues should be adressed in the Discussion section.

If the dogs in this study did not mark as positive any of the patients with respiratory diseases other than COVID-19 this would mean that a very specific VOC or combination of VOCs are characteristic for COVID-19 and should be identified chemically. The other question is if the same VOC or VOCs are characteristic for any new mutations of the virus. This issue should be addressed in the Discussion. Secondly, if the dogs, independently of the experimental design (in vitro and in vivo) and independently of COVID-19 severity, ranging from asymptomatic to pre-symptomatic, sick and very sick patients, alerted with a very high accuracy, the question should be discussed to what they actually alert and what is the origin of the odor and how the odor will be produced.

Minor remarks

L.36 and elsewhere „interrogating” is not a good wording, I suggest „sniffing”

L.36 there are some doubts if sniffing directly the body of patients would be an ideal method because there are some drawbacks e.g. interactions between subjects and dog, fear of dogs, refusal to be sniffed etc .

L.56 it could be expected that dogs that are trained on saliva samples would make more false alerts when sniffing hand palms.

L.103 Here the study objectives and aims should be placed.

L.115 „For maximal output” - this wordng is not clear: working dogs have to be rewarded not for a maximal output but to produce an association between an odor and the reward (operant conditioning)

L.115 „….dogs must be rewarded for each positive finding….” – not quite precise statement: there is also a variable reinforcement ratio – rewording not for every bet only for some of the correct positive findings. This issue concerns real screening under „true” double blind conditions, where the dogs should not be rewarded for each positive finding because it is not known if the alert was correct or false

L.121 Rather „olfactory ability” than „scent power”

L.147 Do the do the dogs identify the virus or alert to the VOCs produced by changed cell metabolisms during infection ?

Tab.1. were some SARS-positive donors asymptomatic ?

L.155 which environmental modifications are meant ?

L.166 – what was the correct behavior (alert), who activated the clicker ?

L.169 how many trials per day ?

L.173 – usually dogs make some false alerts of missess from time to time. How many error-free consecutive trials have to be made?

L.194 what was a criterion for passing to the next training stage (% of correct alerts?)

L.198 The training method for "in vivo” screening was not sufficiently described. How many donors were used for the training? Here it could be understood that 3 donors were used for the training, however in supplementary info page 5 third line from the bottom 400 subjects used for the training are mentioned. What was the first stage of the in vivo training ?

L.201-202 – what kind of samples were collected for the in vivo training (sweat or saliva or secretions) ? were those samples taken only from 51 COVID positive patients, meaning that >100 samples per donor ? How were the samples handled for biosecurity ?

L.207 - dog training was partly described in the paragraph Design and Sample Size.

There is a confusion in describing particular sections: first the animals should be described, then the collecting of odor samples, dog training and statistical methods.

L.226-227 „..only after obtaining very high diagnostic metrics in vitro…”. – how high were these metrics ? >90% ?

L.229 – siffing hand palms without prior washing may involve such issues as confounding odors of food, when a person ate something shortly before being sniffed by the dog, attractive or aversive individual odor of some persons, etc. Also dog-human interaction, fear of dogs may play a role. These issues should be discussed in the revision.

L.320-322 Does the RNA smell? – probably dogs alert to some VOCs produced by cells during infection with SARS-Cov-2. Is there a direct relation between the number of copies of ssRNA/mL and the amount of those putative VOCs ?

L.337 Detailed methodology should be described in the M&M section of the manuscript and not in the Results.

L.341 and 351. In practice it is hardly possible to achieve all trials without any error. Only a series of error-free trials is usually possible. The question is how low was such series – one or 5 or 10 or more trials on 100 samples ?

L.352 – were „zero” trials conducted ? (with no target odor among the 100 negative samples)

L.369-370 „…When the dogs interrogated the first positive patient, all six recognized the scent-print of SARS-CoV-2 and went down without hesitation….” - on videos S1 and S2 played at slow motion some hesitations of dogs can be seen. The first positive patient could be actually alerted without any hesitation but the question is if the further (many) patients would be indicated without any hestitations as well.

L.370-373 „….it proved to be a difficult endeavor…..”. It seems that the training of dogs to alert to SARS-Cov-2 on any belongings of the patients would be too challenging. It would be better to standardize the method and to improve canine proficiency at working on unified odor samples or on people.

L.377 It seems that the confounding effect of individual odor of positive patients tested in vivo plays no role ?

L.397-398 some parts of the description of material and methods are doubled and scattered throughout the main text and the supplementary info. This makes the whole submission too long and difficult to follow.

Figs 5 and 6 seem to be redundant if the same results were given in tables 3 and 4 respectively ?

L.537-542 „…We also observed several times during training that the dogs spontaneously marked as positive the scientists that had touched any COVID-19 patient, or the cell phones of nurses and physicians in care of COVID-19 patients. It means that trained canines detect the scent-print of SARS-CoV-2 in contaminated individuals or in their belongings and, since contamination could lead to infection [39], the dogs actually identify potential COVID-19 cases before infection takes place…” – this statement should be critically revised and clear limitation should be indicated. If the dogs „several times” alerted positively to the experimenters or to the belongings of the hospital staff, there is no proof that the dogs identify COVID-19 before infection takes place. This could be simply false alerts and there are no proofs how many misses (false negative = not detecting cases before infection) would be found.

L.569-573 „…..expert detection dogs remembered a new scent-print 98% of the time as long as it was located first in a line-up with five distractors, but performance went down when the positive stimulus was located farther, dropping to 11.5% at the sixth location [28]. Therefore, it seems impossible for a dog to remember the odor of up to 12 different individuals randomly allocated among 100 distractors….” - the other studies do not support this finding, as the dogs are able to memorize much more than 12 individual scents and the percentage of alerts to memorized scent does not depend of the location in the lineup.

Fig 7. Testing 550 individuals in vivo within 5 hours seems to be definitely too strenuous for dogs. Even testing 110 persons within one hour of work, depending on weather conditions would be very strenuous. It should precised how long was a work bout without break.

In summary, this paper has some merits and would be acceptable for publication in PloS after major revision, considering all critical points and indicating clearly limitations of the metod.

6. PLOS authors have the option to publish the peer review history of their article (what does this mean?). If published, this will include your full peer review and any attached files.

Reviewer #1: No

---

## [Author Response · Author response to Decision Letter 0]

29 Jul 2021

Following the instructions of the Academic Editor, we uploaded a file labeled "Response_to_Reviewers" containing a point-by-point answer to comments from the editor and reviewer.

---

## [Decision Letter · Decision Letter 1]

4 Aug 2021

PONE-D-21-16873R1

Highly sensitive scent-detection of COVID-19 patients in vivo by trained dogs

PLOS ONE

Dear Dr. Vesga,

Thank you for submitting your manuscript to PLOS ONE. After careful consideration, we feel that it has merit but does not fully meet PLOS ONE’s publication criteria as it currently stands. Therefore, we invite you to submit a revised version of the manuscript that addresses the points raised during the review process.

English should be edited as the reviewer suggested.

We look forward to receiving your revised manuscript.

Kind regards,

Etsuro Ito

Academic Editor

PLOS ONE

Journal Requirements:

Reviewers' comments:

Reviewer's Responses to Questions

**Comments to the Author**

1. If the authors have adequately addressed your comments raised in a previous round of review and you feel that this manuscript is now acceptable for publication, you may indicate that here to bypass the “Comments to the Author” section, enter your conflict of interest statement in the “Confidential to Editor” section, and submit your "Accept" recommendation.

Reviewer #1: (No Response)

2. Is the manuscript technically sound, and do the data support the conclusions?

Reviewer #1: Yes

3. Has the statistical analysis been performed appropriately and rigorously? 

Reviewer #1: Yes

4. Have the authors made all data underlying the findings in their manuscript fully available?

Reviewer #1: Yes

5. Is the manuscript presented in an intelligible fashion and written in standard English?

Reviewer #1: No

6. Review Comments to the Author

Reviewer #1: The authors addressed sufficiently my comments in their rebuttal, but have only partly incorporated amendments into the revision. First of all, the structure of the submission should be further improved, since it remains not concise enough and chaotic. Eventually, it is up to the Editor of PloS to decide if the overall structure of the revision is acceptable or should be changed. In my opinion, however, the main body of a paper should contain all essential information and data of the study. I have no comments to the revised Introduction section including hypotheses, except for the lines 141-145 that fit rather to the M&M section than to the Introduction. I think in the Material and Methods the reader should be able to find all essential information on animals, collecting and handling of odor samples, training procedure and statistical methods used in the study, without searching for supporting information. The supporting information is useful for somebody who wants to learn all additional details of the study. In the present form of the revision, the reviewer (and the readers), have to jump to the supporting info and back to the main text, which is inconvenient and makes following of the paper not easy.

Being not an English native speaker, I will not evaluate the language of the paper, however, to me the revision requires extensive editing of English by a native speaker, who is familiar with specific wording concerning both COVID-19 epidemiology and canine detection. Nevertheless, I try to give some suggestions:

L.48-49, I suggest "...A real-life (in vivo) performance was determined 75 days after in vitro effectiveness assay...”

L.51 Here and elsewhere: is the word "interrogation" appropriate in this context ? I would suggest: " ...Three dogs were used to examine the scent of 350 volunteers, who agreed to participate both in test with canines and in rRT PCR testing...”

L.55-58 This statement is imprecise. The task of the dogs is not to discriminate odoriferous contamination from infection since probably all odor samples collected in reality, are to some extent contaminated, and no pure "infection odor" exists. The dog should be rather trained to ignore contamination and to indicate "infection odor" regardless its contamination.

L.76 The nations did not demonstrate. I suggest ".. It was clearly demonstrated in several countries that...."

L.111 I suggest „ … the dog must be reinforced……with a reward……”

L. 115 I suggest „…selection process of dog candidates for the olfactory training e.g. for detection of explosives is needed….”

L.117 „exceeds” instead of „excels”

L.118 „of working in the scent lineup”

L.119 „human odor” instead of „human subject”

L.123. „….learning ability, trainability and ability to cooperate with humans…” instead of „minds”.

L.141-144 – this passage fits rather to the M&M section than to the Introduction.

L.145 „Ultimate goal” instead of „product” (?)

L.155-160 this passage is redundant since it reiterates information given in the Introduction.

L.160-164 (up to „…from each other…”) this passage should be included into Material section (collecting odor samples)

L.164-178 this passage should be shifted to line 211 (Dog training)

L.166. Fig.1. showing the training phases should be cited here.

L.196-200 Should be moved to the Material section - collecting odor samples

L.202-208 A separate paragraph on Ethical permission, should be moved to the beginning of the M&M section. The passage in lines 202-208 has nothing to do with Sample size.

L.235 My suggestion: „ ……could not bite, lick or touch ….”

L.265-270 Statistical analyses should be the last separate paragraph of the M&M section.

L.447-449 Rewarding some but not all correct alerts during the effectiveness assay cannot be considered as human error. In real screening scenario neither the experimenter nor the handler does know if the dog's response was correct or wrong (true double blind procedure). Therefore during real screening scenario, basically no rewarding have to be applied. Increasing the false negative alerts, or false positive alerts as a consequence of not rewarding the dog for EVERY correct alert on real people, is one of the critical points of the canine screening. It remains an open question, if the dogs that are subjected to a sustaining/improving training in order to reduce false negative and false positive alerts, using odor samples and lineup, would equally well alert on live people (in long term). There is something like context dependent olfactory learning that has to be taken into consideration.

L.462 „bred” instead of „created”

L.463 „suitability” instead of „excellence”

L.464 add „concentration ability”

L.465 „individual dogs” instead of „individual prospect”

L.468 „…the canine recognizes variations….” (?) – rather ignores variations and indicates regardless of distracting odors

L.469 „source” is redundant here. I suggest: „indicates the target odor regardless....”

L.470 but is surprising in view of other medical detection dog training e.g. for cancer detection

L.475-478 This sentence should be rewritten since it is confusing. It is not a problem of using urine as both positive (cancer) and negative (healthy stimulus), but the problem of using the same samples (donors) for the training and testing. Also, Ellier et al. 2014 was not the only study that recommended using odor samples from many donors and conducting the training and testing using different samples (donors).

L.480-481 „ …..using the same type of secretion during the foundational training does not favor errorless discrimination learning…..” (1) - what is the foundational training? - perhaps initial training ?, (2) – finally the dogs have to discriminate odor samples from sick vs healthy humans and not sick humans vs sterile saline solution. The dogs may be perfect at what the authors label as „errorless dicrimination training”, but may show poor performance in real screening scenario.

L.483 what stands for Effect size ? perhaps simply "The different diagnostic metrics"?

L.488-490 „…Had they been scenting in search of odors other than their target…” please rewrite to be more clear. Perhaps: „..Had they been alerting to odors other than the target odor…”?

L.490-492 – please rewrite to be more clear

L.500-508 I suggest to delete the passage in lines 500-508 because canine learning as such goes beyond the scope of this study

L.560 while infectious diseases could be detected in seconds, could they also be controlled in seconds?

Fig 1. „…..training phases and experimental design were planned…”? – or were conducted ?

Table 3 The term Effect size may be confusing. I suggest to delete (%) in the first left side column and insert (%) instead of Effect Size in column captions

In my opinion this is an interesting study that is worth publishing, but would definitely benefit from a better preparation of the second revision, including restructuring and extensive editing of English. Some parts of the Discussion are still not clear or difficult to follow. Therefore I recommend minor revision before final acceptance.

7. PLOS authors have the option to publish the peer review history of their article (what does this mean?). If published, this will include your full peer review and any attached files.

Reviewer #1: No

---

## [Author Response · Author response to Decision Letter 1]

27 Aug 2021

Medellin, 27 August 2021

Dear Dr. Ito,

This document contains my point-by-point response to you and to Reviewer 1. We are indebted forever, because thanks to the peer review and editorial process, the manuscript is now readable and understandable.

 In blue font, I left your words and those of R1. In red font, my answers. When citing the new text, I used quotes, but left the new text in black font to ease reading. The last step was to shorten the manuscript, and then it was reviewed by an English-native scientist familiar with the methodology (Tonie E. Rocke, PhD. USGS National Wildlife Health Center). Although all suggestions from R1 were not only welcomed but incorporated in the final version of the manuscript, a few might have been let out (unintentionally) or modified slightly along this process. 

Sincerely,

Omar Vesga, MD.¬¬

¬¬¬¬¬¬¬-________________________________________________________________________________

PONE-D-21-16873R1

Highly sensitive scent-detection of COVID-19 patients in vivo by trained dogs

PLOS ONE

Dear Dr. Vesga,

English should be edited as the reviewer suggested. 

Done. After introducing the changes described below, the manuscript was reviewed and edited (in style) by a US native scientist who is familiar with the methods employed in this study.

Please submit your revised manuscript by Sep 18 2021 11:59PM. 

Done (this document).

Done (uploaded along with this document).

Done (uploaded along with this document).

Etsuro Ito

Academic Editor

PLOS ONE

Journal Requirements:

References checked for completeness and correctness; additional references replaced some cited in the previous version. We found no retracted papers.

Reviewers' comments:

Reviewer's Responses to Questions

Comments to the Author

Reviewer #1: The authors addressed sufficiently my comments in their rebuttal, but have only partly incorporated amendments into the revision. First of all, the structure of the submission should be further improved, since it remains not concise enough and chaotic. Eventually, it is up to the Editor of PloS to decide if the overall structure of the revision is acceptable or should be changed. In my opinion, however, the main body of a paper should contain all essential information and data of the study. I have no comments to the revised Introduction section including hypotheses, except for the lines 141-145 that fit rather to the M&M section than to the Introduction. I think in the Material and Methods the reader should be able to find all essential information on animals, collecting and handling of odor samples, training procedure and statistical methods used in the study, without searching for supporting information. The supporting information is useful for somebody who wants to learn all additional details of the study. In the present form of the revision, the reviewer (and the readers), have to jump to the supporting info and back to the main text, which is inconvenient and makes following of the paper not easy. 

We tried carefully to incorporate all the amendments suggested by R1. When in disagreement (once or twice), we explained our reasons (in this response). 

Being not an English native speaker, I will not evaluate the language of the paper, however, to me the revision requires extensive editing of English by a native speaker, who is familiar with specific wording concerning both COVID-19 epidemiology and canine detection. Nevertheless, I try to give some suggestions. 

The structure was edited by myself first, and then by a native English-speaking scientist, as suggested by R1, trying to make the manuscript concise and clear without sacrificing the substance. It moved the line numbers and I had to write here the text so R1 can check what we did with every one of her/his observations. The Results section was incorporated back to the manuscript. Now, there are no Methods in the Supporting information file.

L.48-49, I suggest "...A real-life (in vivo) performance was determined 75 days after in vitro effectiveness assay...”. 

It was done as suggested, but correcting the expression underlined above. Now, the sentence reads this way: "Seventy-five days after finishing in vivo efficacy experiments, a real-life study (in vivo effectiveness) was executed among the riders of the Metro System of Medellin, deploying the human-canine teams without previous training or announcement.”.

L.51 Here and elsewhere: is the word "interrogation" appropriate in this context ? I would suggest: " ...Three dogs were used to examine the scent of 350 volunteers, who agreed to participate both in test with canines and in rRT PCR testing...” 

It was done as suggested, correcting the expression underlined above. Now, the sentence reads this way: “Three dogs were used to examine the scent of 550 volunteers who agreed to participate both in test with canines and in rRT-PCR testing.”.

L.55-58 This statement is imprecise. The task of the dogs is not to discriminate odoriferous contamination from infection since probably all odor samples collected in reality, are to some extent contaminated, and no pure "infection odor" exists. The dog should be rather trained to ignore contamination and to indicate "infection odor" regardless its contamination. 

The imprecise statement was eliminated from the Abstract. Now it reads this way: “Canine scent-detection in vivo is a highly accurate screening test for COVID-19, and it detects more than 99% of infected individuals independent of key variables, such as disease prevalence, time post-exposure, or presence of symptoms.”. Those variables had been mentioned before, so we just move them down without altering the meaning of the Abstract. 

L.76 The nations did not demonstrate. I suggest ".. It was clearly demonstrated in several countries that....". 

The new line reads this way: “It was clearly demonstrated in several countries that early and massive testing, followed by immediate isolation in designated areas away from home and rigorous contact-tracing, were the only measures that effectively stopped the pandemic even before the first vaccine was available [6].”.

L.111 I suggest „ … the dog must be reinforced……with a reward……” 

The new line reads this way: “Working dogs must be reinforced for each positive finding with a reward that conveys an extremely high value for them, and performance depends heavily on the intensity of the expectations that such reward generates in their brains during foundational training [26].”

L. 115 I suggest „…selection process of dog candidates for the olfactory training e.g. for detection of explosives is needed….” 

The new line reads this way: “Even within optimal training conditions, not all canine individuals will give their best to gain a reward, and a rigorous selection process of dog candidates for the olfactory training is needed [27].”.

L.117 „exceeds” instead of „excels”. 

The new line reads this way: “Bloodhound is a working breed with exceptional olfactory ability,”. 

L.118 „of working in the scent lineup”. 

The new line reads this way: “…but is not suitable for medical detection because most individuals do not enjoy working in the scent lineup.”.

L.119 „human odor” instead of „human subject”. 

Line 119 was eliminated.

L.123. „….learning ability, trainability and ability to cooperate with humans…” instead of „minds”. 

The new line reads this way: “but cognition research is providing solid evidence that dogs indeed have unusual learning ability compared with other nonhuman animals…”.

L.141-144 – this passage fits rather to the M&M section than to the Introduction. 

It is the last paragraph of the introduction, and as such it has the purpose of preparing the reader for the methods and results. It is the usual style in most journals focused on human medical sciences, including PLoS ONE, which states this in their Submission Guidelines under the topic “Introduction”: 

The introduction should:

• Provide background that puts the manuscript into context and allows readers outside the field to understand the purpose and significance of the study

• Define the problem addressed and why it is important

• Include a brief review of the key literature

• Note any relevant controversies or disagreements in the field

• Conclude with a brief statement of the overall aim of the work and a comment about whether that aim was achieved

L.145 „Ultimate goal” instead of „product” (?). 

The intention of the sentence is to confirme that the aim was achieved. Therefore, there is no place to add an additional goal when all five objectives had been mentioned in the previous paragraph of the introduction. Since the noun “product” (a thing produced by labor) is what R1 objects, we changed it for “outcome” (a final product or end result; a conclusion reached through a process of logical thinking). The fact that the PPV dropped during the effectiveness assay does not invalidate efficacy data, therefore it is not misleading to say that we ended up with a “very fast and cost-effective screening method”. 

L.155-160 this passage is redundant since it reiterates information given in the Introduction. 

The sentence was replaced by this: “Fig 1 shows the training program and experimental design.”. 

L.160-164 (up to „…from each other…”) this passage should be included into Material section (collecting odor samples). 

A new sub section called “Specimen collection for in vitro work” was created in Methods, between Sample Size and Dog Training. The paragraph in lines 160-164 was moved there.

L.164-178 this passage should be shifted to line 211 (Dog training). 

Done.

L.166. Fig.1. showing the training phases should be cited here. 

Done.

L.196-200 Should be moved to the Material section - collecting odor samples. 

Done.

L.202-208 A separate paragraph on Ethical permission, should be moved to the beginning of the M&M section. The passage in lines 202-208 has nothing to do with Sample size. 

Done.

L.235 My suggestion: „ ……could not bite, lick or touch ….”. 

The new line reads this way: “…and the difference between both was that animals allocated to experimental groups could not bite, lick, or touch D1 or D2,…”.

L.265-270 Statistical analyses should be the last separate paragraph of the M&M section. 

Done.

L.447-449 Rewarding some but not all correct alerts during the effectiveness assay cannot be considered as human error. In real screening scenario neither the experimenter nor the handler does know if the dog's response was correct or wrong (true double blind procedure). Therefore during real screening scenario, basically no rewarding have to be applied. Increasing the false negative alerts, or false positive alerts as a consequence of not rewarding the dog for EVERY correct alert on real people, is one of the critical points of the canine screening. It remains an open question, if the dogs that are subjected to a sustaining/improving training in order to reduce false negative and false positive alerts, using odor samples and lineup, would equally well alert on live people (in long term). There is something like context dependent olfactory learning that has to be taken into consideration. 

The expression “human error” was taken off the sentence, which now reads this way: “. One explanation was handlers rewarding some but not all alerts during the effectiveness assay but, in fact, rRT-PCR results showed that many correct alerts passed unrewarded; it confused the dogs and caused even more false negative alerts.”.

L.462 „bred” instead of „created”. 

The new line reads this way: “Second, all six dogs were successful as medical detectors despite belonging to breeds not intended specifically for scent-detection.”:

L.463 „suitability” instead of „excellence”. 

See answer to L.465 below.

L.464 add „concentration ability”. 

See answer to L.465 below.

L.465 „individual dogs” instead of „individual prospect”. 

The new line reads this way: “It supports recent data showing that, more than the breed, the best predictors of suitability for medical detection dogs are the levels of motivation, stamina, determination, resilience, and concentration ability of the individual dog [42,43].”:

L.468 „…the canine recognizes variations….” (?) – rather ignores variations and indicates regardless of distracting odors. 

The new line reads this way: “This process, called generalization, applies to learning theory, and in reference to scent-detection means that the canine ignores variations of the positive stimulus and indicates its source regardless of distracting odors [44].”.

L.469 „source” is redundant here. I suggest: „indicates the target odor regardless....”. 

Actually, we trained our dogs to alert on the SOURCE of the target scent instead of the odor itself. It is useful for outdoors scent-work, where a moving odor causes the dog could alert before arriving to the source. On the scent-line with humans, it could cause the dog to alert on the wrong individual when wind is blowing towards the dog.

L.470 but is surprising in view of other medical detection dog training e.g. for cancer detection. 

The word “surprising” was eliminated. The new sentence reads this way: “Generalization after exposure to just three specimens is to be expected if errorless learning principles are the foundations of training [31].”.

L.475-478 This sentence should be rewritten since it is confusing. It is not a problem of using urine as both positive (cancer) and negative (healthy stimulus), but the problem of using the same samples (donors) for the training and testing. Also, Ellier et al. 2014 was not the only study that recommended using odor samples from many donors and conducting the training and testing using different samples (donors). 

The new sentence reads this way: “However, experimental evidence in favor of such hypothesis is scarce, and most citations refer to a work in which urine was employed as positive and negative stimulus during training [48].”.

L.480-481 „ …..using the same type of secretion during the foundational training does not favor errorless discrimination learning…..” (1) - what is the foundational training? - perhaps initial training ?, (2) – finally the dogs have to discriminate odor samples from sick vs healthy humans and not sick humans vs sterile saline solution. The dogs may be perfect at what the authors label as „errorless dicrimination training”, but may show poor performance in real screening scenario. 

True, but in vivo experiments (efficacy trial, phase 3) demonstrated a very high effect size for every diagnostic metrics, including PPV. We did not invent errorless discrimination learning, it is actually the product of a very solid scientific work, as cited in the manuscript. The fact the dogs failed during the effectiveness assay does not affect the truth of errorless learning because the reward system was changed. To eliminate the term “foundational training” the new sentence reads this way: “Beyond the many variables specific to certain diseases and specimens that might be responsible for a greater level of difficulty for the dog, using the same type of secretion when the dogs are first trained for scent-detection does not favor errorless discrimination learning [31,32,45].”.

L.483 what stands for Effect size ? perhaps simply "The different diagnostic metrics"? 

Just to answer the question of R1: effect size is a mathematical term that refers to the magnitude of a difference between two groups or, simply put, the most important result of most experiments. In this study, the effect size is the difference in diagnostic performance between the index test (K9 scent-detection) and the gold standard (rRT-PCR). To validate our dx test, we used these metrics, or effect sizes: SEN, SPC, PPV, NPV, ACC, and LR. Reviewer 1 is pointing correctly that we have created confusion by calling the values of these metrics “effect sizes”, because one thing is the name of a parameter or a metric or an effect size, and other is its numerical value or magnitude. In the new version, all references to the numeric result of any metric (aka effect size) are called “value”, “magnitude”, or “mean”. 

L.488-490 „…Had they been scenting in search of odors other than their target…” please rewrite to be more clear. Perhaps: „..Had they been alerting to odors other than the target odor…”? 

The new sentence reads this way: “Had they been alerting to odors other than their target odor, SPC would have been very low, and it was also >95%.”.

L.490-492 – please rewrite to be more clear. 

The new sentence reads this way: “While SEN and SPC refer to the index test (canine scent-detection), predictive values quantify the probability that the participants truly had or not COVID-19, taking the reference standard (rRT-PCR) as the truth.”.

L.500-508 I suggest to delete the passage in lines 500-508 because canine learning as such goes beyond the scope of this study. 

Done. The reader is directed to references 49-53 in lines 497-499: “Previous knowledge of dog behavior and cognition also suggest that our dogs were not relying in memory when they detected SARS-CoV-2 and discriminated it among so many other odoriferous cues [44,49-54].”. 

L.560 while infectious diseases could be detected in seconds, could they also be controlled in seconds? 

It is if, as in South Korea and New Zealand, positive patients are isolated immediately. The chain of transmission stops right there. That said, we understand that R1 is pointing to the fact that the data had nothing to do with controlling the transmission of COVID-19. Therefore, the words “and control” were eliminated. The new sentence reads this way: “With improved training methods, canines could, in the near future, provide a sensitive and effective method to detect infectious diseases in a matter of seconds.”.

Fig 1. „…..training phases and experimental design were planned…”? – or were conducted ?. 

The new title of Fig 1 reads this way: Fig 1. Efficacy studies. Flow chart depicting the order in which training phases and experimental design were conducted.

Table 3 The term Effect size may be confusing. I suggest to delete (%) in the first left side column and insert (%) instead of Effect Size in column captions. 

Done. The title of all columns was modified to reflect the difference between the effect size (metric) and its magnitude (numerical value). 

In my opinion this is an interesting study that is worth publishing, but would definitely benefit from a better preparation of the second revision, including restructuring and extensive editing of English. Some parts of the Discussion are still not clear or difficult to follow. Therefore I recommend minor revision before final acceptance. 

Done. After introducing the changes described above, the manuscript was reviewed and edited (in style) by a US native scientist who is familiar with the methods employed in this study.

---

## [Editor Report · Decision Letter 2]

2 Sep 2021

Highly sensitive scent-detection of COVID-19 patients in vivo by trained dogs

PONE-D-21-16873R2

Dear Dr. Vesga,

We’re pleased to inform you that your manuscript has been judged scientifically suitable for publication and will be formally accepted for publication once it meets all outstanding technical requirements.

Kind regards,

Etsuro Ito

Academic Editor

PLOS ONE

---

## [Editor Report · Acceptance letter]

14 Sep 2021

PONE-D-21-16873R2 

Highly sensitive scent-detection of COVID-19 patients in vivo by trained dogs 

Dear Dr. Vesga:

I'm pleased to inform you that your manuscript has been deemed suitable for publication in PLOS ONE. Congratulations! Your manuscript is now with our production department. 

Kind regards, 

on behalf of

Prof. Etsuro Ito 

Academic Editor

PLOS ONE